# Mature oligodendrocytes bordering lesions limit demyelination and favor myelin repair via heparan sulfate production

Magali Macchi[1†], Karine Magalon[1†], Céline Zimmer[1], Elitsa Peeva[2], Bilal El Waly[1], Béatrice Brousse[1], Sarah Jaekel[2], Kay Grobe[3], Friedemann Kiefer[4], Anna Williams[2], Myriam Cayre[1], Pascale Durbec[1]*

[1]Aix Marseille Univ, CNRS, IBDM, Marseille, France; [2]MRC Centre for Regenerative Medicine, Multiple Sclerosis Society Centre for Translational Research, University of Edinburgh, Edinburgh, United Kingdom; [3]Institute of Physiological Chemistry and Pathobiochemistry and Cells-in-Motion Cluster of Excellence (EXC1003-CiM), University of Münster, Münster, Germany; [4]Max Planck Institute for Molecular Biomedicine, Münster, Germany

**Abstract** Myelin destruction is followed by resident glia activation and mobilization of endogenous progenitors (OPC) which participate in myelin repair. Here we show that in response to demyelination, mature oligodendrocytes (OLG) bordering the lesion express Ndst1, a key enzyme for heparan sulfates (HS) synthesis. Ndst1+ OLG form a belt that demarcates lesioned from intact white matter. Mice with selective inactivation of Ndst1 in the OLG lineage display increased lesion size, sustained microglia and OPC reactivity. HS production around the lesion allows Sonic hedgehog (Shh) binding and favors the local enrichment of this morphogen involved in myelin regeneration. In MS patients, Ndst1 is also found overexpressed in oligodendroglia and the number of Ndst1-expressing oligodendroglia is inversely correlated with lesion size and positively correlated with remyelination potential. Our study suggests that mature OLG surrounding demyelinated lesions are not passive witnesses but contribute to protection and regeneration by producing HS.

*For correspondence:
pascale.durbec@univ-amu.fr

[†]These authors contributed equally to this work

**Competing interests:** The authors declare that no competing interests exist.

## Introduction

In multiple sclerosis (MS), OLG are the target of inflammatory and immune attacks and their death results in multiple focal demyelinated lesions in the CNS. Myelin loss remains in defined areas, rather than expanding to involve all of the white matter, and the mechanism by which demyelinated lesions stop expanding is not understood. Some of these lesions may stop expanding as inflammation is controlled and lesions are remyelinated. Remyelination involves OPC recruitment to the lesion, differentiation into myelin forming cells and remyelination of denuded axons, and the success of this depends on environmental context, including secreted factors from neighboring cells (*Franklin and Ffrench-Constant, 2008*; *Patrikios et al., 2006*). This repair process, which occurs spontaneously in MS patients, is highly variable between patients and between lesions (*Patrikios et al., 2006*). Regenerative failure is mainly attributed to defects in OPC recruitment towards the demyelinated areas (*Boyd et al., 2013*) and/or to their incapacity to differentiate into myelinating OLG at the lesion site (*Franklin and Ffrench-Constant, 2008*; *Wolswijk, 1998*; *Chang et al., 2002*; *Franklin, 2002*).

Multiple factors are involved in this regenerative process including those produced by reactive astrocytes or microglia and macrophages (*Miron et al., 2013*). These contribute to myelin

destruction, but also to myelin debris removal and beneficial effects by secreting factors that directly or indirectly support remyelination (*El Waly et al., 2014*; *Emery, 2010*; *Aguirre et al., 2007*; *Williams et al., 2007*; *Courtès et al., 2011*; *Vernerey et al., 2013*). Interestingly oligodendrocyte lineage cells also produce factors that modulate remyelination, such as the morphogen shh which is produced by OLG and OPC at the onset of demyelination in lysophosphatidyl choline (LPC)-induced lesions (*Ferent et al., 2013*). In this context, blocking Shh activity induces an increase in lesion size and a block in OPC proliferation and differentiation, and conversely Shh overexpression leads to the attenuation of the lesion extent and promotes oligodendrogenesis (*Ferent et al., 2013*). Identifying such actors involved in myelin damage and remyelination is needed for the design of future protective and regenerative therapies.

The presence of a multitude of signals regulating specific steps of remyelination raises the hypothesis that key factors may be necessary to integrate all these cues. One of these key factors may be HS proteoglycans (HSPG), as there is now compelling evidence that HSPGs play a critical role in regulating spatiotemporal coordination of signals in the extracellular microenvironment of many tissues during brain development and in adulthood (*Sarrazin et al., 2011*). HS chains consist of linear repeated disaccharide units of N-acetyl glucosamine and glucuronic acid which are synthesized on proteoglycan core proteins. Ndst enzymes perform the first step of these sugar modifications thus specifying the functional properties of HSPGs (*Lindahl et al., 1998*; *Perrimon and Bernfield, 2000*; *Carlsson et al., 2008*). Among the four known Ndst enzymes, Ndst1 appears as the key enzyme for addition of N-sulfated motifs to HS chains in brain during development, as shown by limited functional redundancy mediated by other Ndst enzymes (2-4) in *Ndst1* KO mice (*Grobe, 2005*; *Pallerla et al., 2007*). During development, HS proteoglycans provide an important signaling scaffold allowing spatial concentration or trapping of numerous molecules such as morphogens and growth factors (*Matsuo and Kimura-Yoshida, 2014*) and the control of receptor activity (*Matsuo and Kimura-Yoshida, 2014*; *Gallagher, 2001*; *Häcker et al., 2005*; *Parker and Kohler, 2010*). Following CNS injury, HSPGs are known to play a pivotal role in post-lesional plasticity and regeneration (*Iseki et al., 2002*; *Hagino et al., 2003*). Some HS proteoglycans are over-expressed by reactive astrocytes in injured mouse brain and provide positive (*Iseki et al., 2002*) or negative (*Hagino et al., 2003*) environmental support for axon regenerative responses. In vitro, HS proteoglycans can prevent OLG differentiation, maintaining OPC in an immature proliferative phenotype by acting as a FGF-2 co-receptor (*McKinnon et al., 1990*; *Bansal and Pfeiffer, 1997*). Therefore, we hypothesized that HS proteoglycans play an organizing role in controlling myelin damage and repair.

Here we show that mature OLG bordering a demyelinated lesion limit lesion extension and influence OPC mobilization via HS production. Using a model of acute focal demyelination of the corpus callosum in mice, we show that *Ndst1* expression is induced in OLG around the lesion throughout the phases of demyelination and remyelination. *Ndst1* expression and subsequent HS accumulation mostly accumulate at the margin of the lesion, delimiting the lesion from the intact corpus callosum during demyelination. To evaluate the relevance of Ndst1 induction for lesion formation and repair, we exposed genetically modified mice with selective deletion of *Ndst1* in oligodendroglia to focal demyelination of the corpus callosum. Lack of Ndst1 in OLG resulted in an increased lesion size, and a sustained OPC and microglia/macrophage activation at the early stage of remyelination. HS enrichment correlates with and is necessary for the binding around the lesion site of the morphogen Shh, suggesting that Ndst1 expression and HS secretion by OLG enhances Shh signaling after demyelination, thus favoring remyelination (*Ferent et al., 2013*; *Zakaria et al., 2019*). Furthermore, NDST1 expression in OLG was also increased in human postmortem tissues from multiple sclerosis patients. This increased density of NDST1+ OLG in lesions was inversely correlated with the size of the lesion and positively correlated with remyelination.

## Results

### Demyelination triggers *Ndst1* up-regulation by OLG and creates a transient N-sulfated belt around the lesion

To identify candidates that could regulate interactions between progenitors and the injured environment, a microarray analysis was performed to compare gene expression in purified oligodendroglia

from adult healthy and demyelinated animals (*Cayre et al., 2013*). One of the most robustly and significantly up-regulated genes after demyelination was *Ndst1*, a key enzyme of HS proteoglycan synthesis (fold increase of 48.9 and 14.0 in two different trials; $p \le 0.001$; microarray data are available at GEO with accession number GSE47486). This up-regulation of *Ndst1* was confirmed in vivo at 21 days in mice exposed to EAE by in situ hybridization combined with Olig2 labeling, a pan OLG marker. While *Ndst1* was not detected in the corpus callosum of control brains (*Figure 1—figure supplement 1A*), it was highly expressed by the Olig2+ population after EAE in the corpus callosum (*Figure 1—figure supplement 1B–C*) in close proximity to lesion sites (*Figure 1—figure supplement 1C*).

To characterize the up-regulation of *Ndst1* after demyelination, we used LPC to trigger focal demyelination lesions in the mouse corpus callosum (*Figure 1A*). In this model, demyelination is not T cell driven, and demyelination and remyelination proceed in a stereotypic sequence: demyelination occurs within few days, endogenous progenitor mobilization peaks at eight dpi and is followed by OPC differentiation (*El Waly et al., 2014*). Production of new myelin is then observed after 2 weeks. Demyelination of the corpus callosum is clearly visible after LPC injection in a reporter mouse where myelin fluoresces green (*plp-GFP*) (*Spassky et al., 2001*; *Le Bras et al., 2005*) by the total loss of GFP signal around the injection site (*Figure 1—figure supplement 2*). The lesion is also characterized by a strong increase in cell density (due to glia proliferation and to microglia/macrophage infiltration) observed by Hoechst staining (*Figure 1—figure supplement 2*) that strictly coincides with loss of GFP fluorescence.

We first evaluated *Ndst1* expression levels by performing RT-qPCR analysis using the corpus callosum of healthy or demyelinated mice on the ipsi- and contralateral sides to the LPC injection, 8 days post injection (dpi). We quantified a mean 41% increase in the *Ndst1* expression level in the demyelinated corpus callosum compared to healthy corpus callosum (*Figure 1B*; p=0.05). *Ndst1* transcripts were found up-regulated in demyelinated corpus callosum by in situ hybridization at five dpi during demyelination (*Figure 1C–F*), eight dpi (*Figure 1G*), and 14 dpi (*Figure 1H*). At days 5 and 8 dpi, *Ndst1* expressing cells delimited a belt around the lesion site. Weak staining was observed distal to the lesion, in the contralateral side of the corpus callosum and at the core of the lesion (*Figure 1C–D*). Thus, the induction of demyelination in the corpus callosum triggers *Ndst1* up-regulation, and this change is sustained throughout the phases of demyelination and remyelination.

Since Ndst1 catalyzes a mandatory step in the synthesis of HS chains, its expression is likely to reflect the distribution of HS. We thus examined the outcome of Ndst1 activity by analyzing the distribution of N-sulfated motifs after demyelination. To do so, we used the anti-HS antibody 10E4 (*Grobe, 2005*; *Pan et al., 2014*), which exclusively detects the N-sulfated motifs produced by Ndst1 activity (*Pallerla et al., 2007*). While no staining was detected in the contralateral corpus callosum (*Figure 2A*), N-sulfated HS positive puncta formed a belt around the demyelination lesion (*Figure 2B–C*), similar to the profile of Ndst1-induction. Furthermore, as with Ndst1 expression, no HS staining was detected within the lesion (*Figure 2B–C*). This staining is specific as it was absent after treatment with Heparinase, an enzyme that digests N-sulfated motifs on the HS chains (*David et al., 1992*; *Figure 2D*). This correlation between *Ndst1* up-regulation and the presence of a highly N-sulfated microenvironment indicates the functional activity of Ndst1 after LPC-induced demyelination in the corpus callosum.

The phenotype of *Ndst1* expressing cells around the demyelinated lesion was examined at five dpi, using co-staining for several markers combined with *Ndst1* in situ hybridization. We found that *Ndst1* expressing cells are immunopositive for Olig2 (*Figure 3A*) and *Plp*+ using double in situ hybridization (*Figure 3B*). We quantified that 98.0 ± 1.5% *Ndst1*+ cells express Olig2 indicating that virtually all *Nsdt1* cells belong to the oligodendroglia lineage (*Figure 3—source data 1*). We found that Ndst1 expressing cells are immunopositive for Olig2, PDGFRα and CC1 (*Figure 3A,C,D*) and Plp+ using double in situ hybridization (*Figure 3B*) At five dpi, 97.5 ± 1.7% of *Ndst1* expressing cells were co-labeled with CC1, a mature OLG marker (*Figure 3D*), while only 0.8 ± 0.3% co-expressed the OPC marker PDGFRα (*Figure 3C*).

Of note, in the belt delimited by *Ndst1* staining surrounding the lesion, only half of the Olig2+ cells expressed *Ndst1* (45.8 ± 3.4%). This may indicate that not all stages of oligodendroglial maturation or not all oligodendrocytes respond equally to the lesion. We observed that, 8.5 ± 3.2% of PDGFRα+ cells and 48.9 ± 8.5% of CC1+ cells surrounding the lesion co-labeled with *Ndst1*. Our

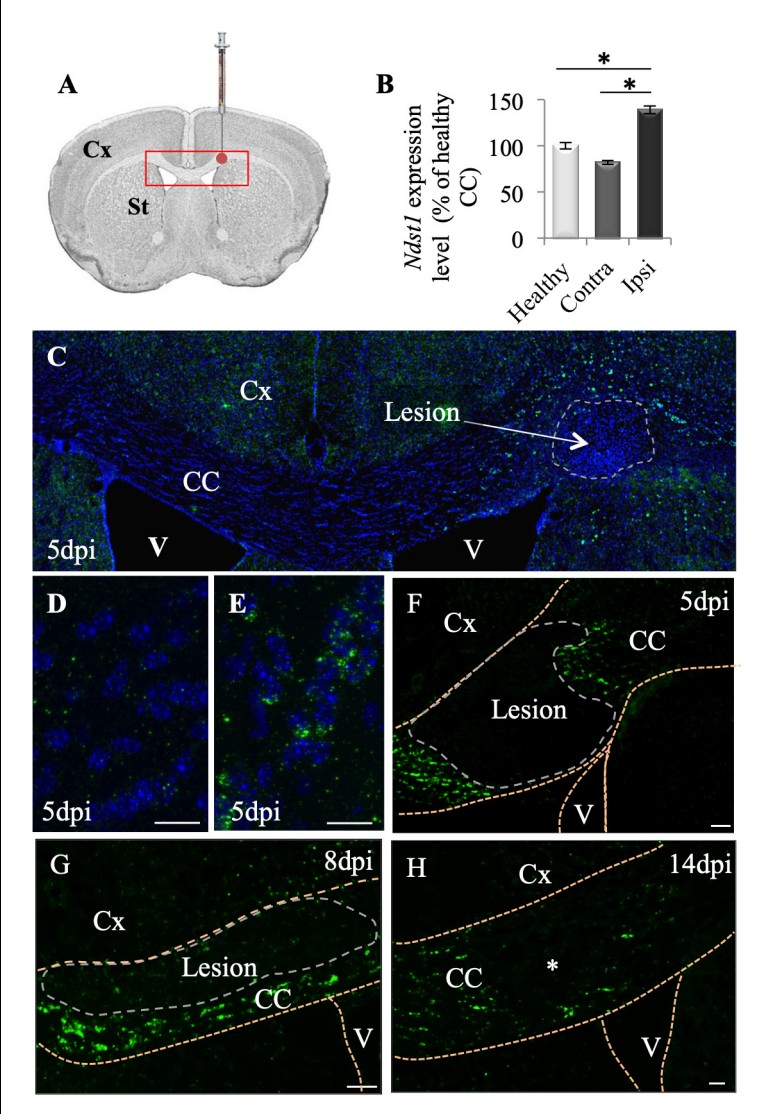

**Figure 1.** *Ndst1* up-regulation upon LPC-induced demyelination of the corpus callosum. (A) Scheme showing the site of LPC injection (red point) in the adult corpus callosum and the location of picture shown in C (red rectangle). (B) *Ndst1* expression levels (RT-qPCR) in the corpus callosum of healthy or demyelinated mice, contralateral (contra) and ipsilateral (ipsi) to the lesion site showing the *Ndst1* up-regulation in the ipsilateral side. Tissues from five mice were pooled in each condition. Error bars represent S.E.M. *p<0.05, non-parametric ANOVA followed by Kruskal-Wallis test (independent two group comparisons). (C–H) *Ndst1* in situ hybridization performed at 5 (C-F, n = 4), 8 (G, n = 4) and 14 (H, n = 4) dpi illustrating the *Ndst1* expression pattern at different time points of demyelination (C–F) and remyelination (G–H). (D–E) Enlarged views of the CC in (C) corresponding to contralateral side (D) and positive cells at the margin of the demyelinated area at the site of LPC injection (E). CC, corpus callosum; Cx, cortex; SVZ, sub-ventricular zone; V, ventricle (structures are delineated by brown dotted lines, lesion with white dotted lines). Scale bars: 50 μm in F, G and H; 20 μm in D, F, H; 10 μm in D and E Asterisk in G indicates the site of injection since the demyelinated lesion is no longer visible at 14 dpi.

The online version of this article includes the following figure supplement(s) for figure 1:

**Figure supplement 1.** *Ndst1* is up-regulated by the Olig2+ cell population in close proximity to inflammation sites in corpus callosum, in the experimental autoimmune encephalomyelitis mouse model of demyelination.

**Figure supplement 2.** *PlpGFP* mice (n = 3) were used to detect demyelinated lesions (A, B, D, E).

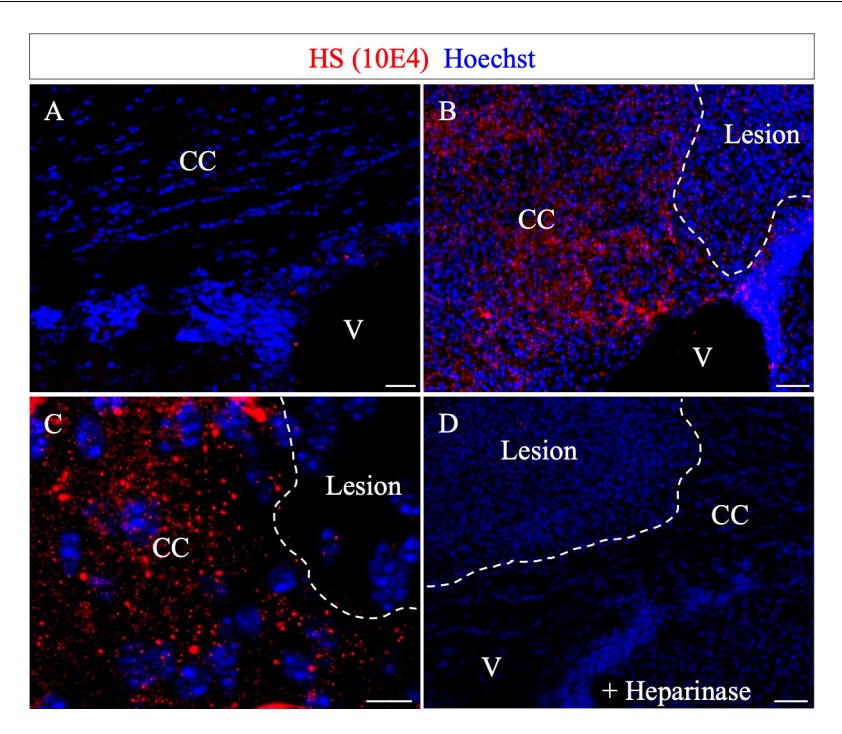

**Figure 2.** N-sulfate-enriched microenvironment forms a belt around the demyelinated lesion. HS (10E4) labeling on the contra- (**A**) and ipsi- (**B–C**) lateral side to the lesion illustrates the generation of a N-sulfated microenvironment surrounding the lesion (delimited by white dashed lines) at five dpi (n = 3). No immunoreactivity was found after Heparinase I treatment (**D**) thus validating the 10E4 antibody specificity. Scale bars: 20 μm in A, B, D; 10 μm in C. CC, corpus callosum; V, ventricle.

data indicate that the majority of Ndst1+ cells around the lesion are mature OLG which are more prone than OPC to activate Ndst1 expression in response to demyelination.

## Deletion of *Ndst1* in the Olig2+ population transiently worsens the extent of demyelination and modifies OPC reactivity after LPC injection

To test if Ndst1 activity in oligodendroglia controls demyelination and/or remyelination, we generated transgenic mice with a conditional deletion of *Ndst1* in Olig2+ cells, by breeding *Olig2-Cre+/-* mice and *Ndst1* $^{Flox/Flox}$ mice (*Grobe, 2005*; *Dessaud et al., 2007*). The efficiency of inactivation of *Ndst1* expression in Olig2 cells was monitored by in situ hybridization in the context of LPC-induced demyelinating lesion, revealing a drastic decrease in *Ndst1* expression in lesioned mutants compared to control (*Figure 4A–F*). As revealed by immunostaining using the anti-HS antibody, a significant reduction of 53% of the staining in N-sulfated HS positive puncta around the demyelination lesion was also observed in mutant compared to control (*Figure 4G–I*; p=0.05). In healthy mice, quantitative analysis of myelin content (*Figure 4—figure supplement 1A–C*; p=0.2), astrocyte density (*Figure 4—figure supplement 1D–F*; p=0.4) and oligodendroglial lineage cell density (*Figure 4—figure supplement 1G–L*; p=0.6, 0.2 and 0.2 for Olig2, CC1 and PDGFRα cell density respectively) revealed no difference between control (*Olig2-Cre+/-*) and mutant (*Olig2-Cre+/-; Ndst1* $^{Flox/Flox}$) adult mice, thus indicating that conditional deletion of *Ndst1* in the Olig2+ cell population does not interfere with brain development and with subsequent myelin maturation.

We first performed LPC-induced demyelination of the corpus callosum in control and mutant mice and measured the size of the lesion at 4, 8 and 14 dpi. As before, lesions were identified based on the high density of nuclei at the injection point (*Figure 1—figure supplement 2*). While no difference was detected at four dpi (0.226 ± 0.036 vs. 0.159±0.033 mm$^3$ in control and mutant mice, respectively; p=0.23), a significant two-fold increase in lesion size was observed in mutant compared to control mice at eight dpi (0.199 ± 0.032 vs. 0.097±0.022 mm$^3$, p=0.023) (*Figure 5A–C*). During

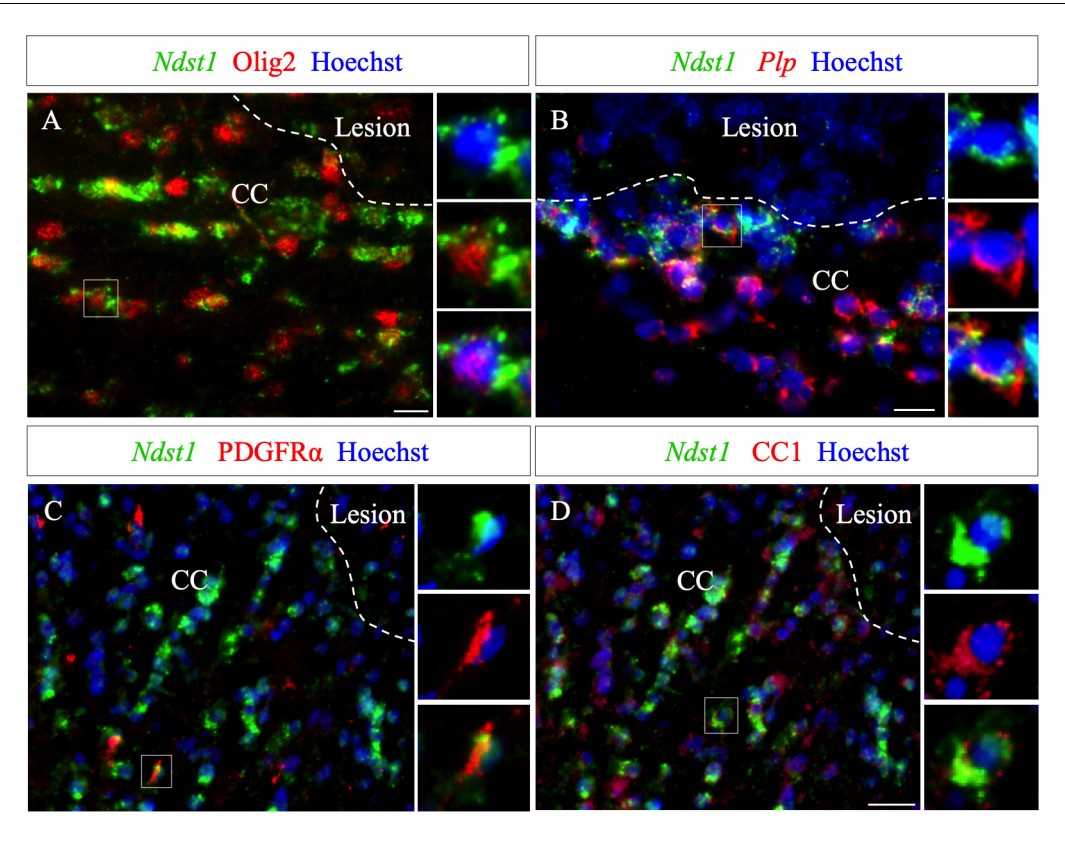

**Figure 3.** Ndst1 expressing cells around the lesion belong to the oligodendroglial lineage. (A–B) *Ndst1* in situ hybridization successively combined with Olig2 immunostaining (A) or *Plp* in situ hybridization (B) labeling, two OLG markers, illustrating *Ndst1* up-regulation in oligodendroglia lineage cells surrounding the lesion site at five dpi (n = 3). (C–D) Representative images of *Ndst1*/PDGFRα (C) and *Ndst1*/CC1 (D) co-labeling illustrating that both OPC (C) and mature OLG (D) up-regulate *Ndst1* after demyelination at five dpi (n = 4). Inserts in (A–D) illustrate boxed regions at high magnification. Scale bars: 20 μm.

The online version of this article includes the following source data for figure 3:

**Source data 1.** Source data files of quantitative analysis of Ndst1 expressing cells around demyelination at five dpi with co-labeled with Olig2, CC1 and PDGFRα.

the remyelination phase (between 8 and 14 dpi), the lesion area decreased in both groups reaching comparable sizes at 14 dpi (0.033 ± 0.02 vs. 0.028±0.012 mm$^3$ in control and mutant mice, respectively; p=0.97) (*Figure 5C*).

We examined how these changes in the local environment in these mutant mice affect OPC mobilization during remyelination by analyzing Olig2+ cells density (*Figure 5D–F*), maturation status (*Figure 5G–I*) and proliferation (*Figure 5J–O*). In accordance with demyelination, there was a marked decrease in Olig2+ cell density within the demyelinated area compared to healthy corpus callosum in both groups at four dpi (55.5 ± 3.3% and 57.3 ± 3.9% decrease in control and mutant mice respectively, p=0.03 and p=0.001) (*Figure 5F*), reflecting the loss of oligodendrocytes. We observe that in both conditions the density of Olig2+ cells returned to uninjected control values at 14 dpi. Quantification of mean cell densities of mature OLG (CC1+) (*Figure 5G–I*) within the lesion throughout the time course revealed no significant difference between the two groups indicating that cell differentiation is not affected by *Ndst1* inactivation.

We observed that the density of Ki67+ proliferating cells within the lesion area was two-fold increased in *Ndst1* mutant mice compared to control mice at eight dpi (217.8 ± 42.8% vs. 100±13.5% in mutant and control mice, respectively, p=0.037) (*Figure 5J–K*). Some of these proliferative cells are OPC since they co-express Olig2 and Ki67 (*Figure 5M–O*). At four dpi, the percentage of proliferating Olig2+ cells was lower (but not significantly different) in mutant mice compared to

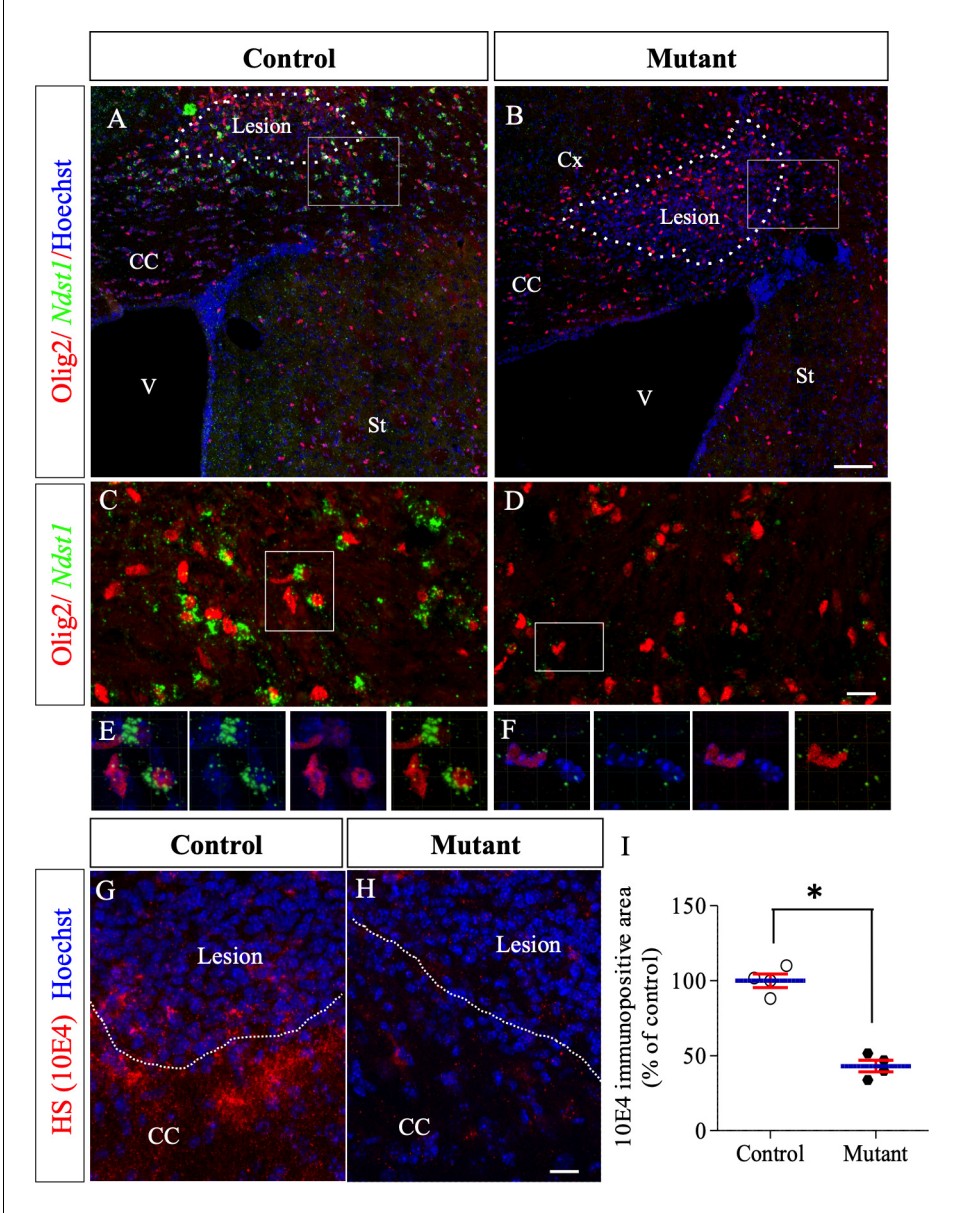

**Figure 4.** Ndst1 inactivation in oligodendrocyte lineage cells in *Olig2-Cre+/-; Ndst1 Flox/Flox* mice. (**A–B**)
Representative images of the lesion site (delineated by white dashed lines) in the corpus callosum of control (**A**)
(n = 2) and mutant (**B**) (n = 2) mice at eight dpi illustrating the enlargement of the lesion size in mutant mice
compared to control mice. Olig2 (in red) is used to label oligodendrocyte lineage cells. In situ hybridization
revealed a marked reduction in *Ndst1* expression surrounding the lesion site in mice with conditional inactivation
in the oligodendroglial lineage cells (**B, D, F**) compared to control mice (**A, C, E**). C and D are high magnifications
of the squares in A and B respectively. E and F are high magnifications of the squares in C and D respectively.
Representative images of 10E4 immunostaining at the lesion site (delineated by white dashed lines. 8dpi) in the
corpus callosum of control (**G**) and mutant (**H**) mice showing a strong reduction of heparan sulfate labeling in
absence of Ndst1 in oligodendrocytes. (**I**) Quantitative analysis of heparan sulfate labeling area fraction in control
and Mutant conditions (n = 4 mice per condition). Error bars represent S.E.M. *p<0.05, non-parametric Mann-
Whitney test (independent two group comparisons). CC, corpus callosum, V, ventricle, St, Striatum. Scale bars: 100
µm in A-; 20 µm in C-D. 30 µm in G-H. Source files of the quantitative analyses are available in the *Figure 4—
source data 1*.

The online version of this article includes the following source data and figure supplement(s) for figure 4:

**Source data 1.** Source data for graph in panel I.
**Figure supplement 1.** Myelin content and glial density in adult unlesioned *Olig2-Cre+/-; Ndst1 Flox/Flox* mice.

*Figure 4 continued on next page*

*Figure 4 continued*

**Figure supplement 1—source data 1.** Source data for graphs in panels C, F, I and L.

control (23 ± 3.6% vs. 17 ± 2.6% of Ki67+/Olig2+ cells in control and mutant mice, respectively, p=0.29) (*Figure 5O*). At eight dpi, the percentage of proliferating Olig2+ cells was significantly increased in mutant mice (7.6 ± 0.8% vs 3.2 ± 0.5% in mutant and control mice, respectively, p=0.0004) indicating a prolongation of OPC reactivity during the repair phase. These data show that OPC reactivity is altered in absence of Ndst1 at the onset of remyelination (8dpi). To note, no significant difference in cell death was detectable in the lesion between the two groups at four dpi (341.7 ± 4.5 and 357 ± 111.1 caspase3+ cells per mm$^2$ in control and mutant mice, respectively, p=0.28).

Together, these results suggest that *Ndst1* expression in the Olig2+ population has no effect on initial demyelination (equivalent lesion size at four dpi) but protects the lesion from enlarging and participates in the control of OPC mobilization.

## Deletion of *Ndst1* in the Olig2+ population modulates microglia/macrophage activation

While the total number of proliferating cells within the lesion area was strongly increased in *Ndst1* mutant mice compared to control mice at eight dpi (*Figure 5J–L*), the percentage of OPC among these cells represent only 7.6% in the mutant. These data suggest that *Ndst1* loss in the Olig2 population indirectly modulates proliferation of surrounding cell types in the context of a demyelinating lesion. To address this, we evaluated the proliferation and activation states of the macrophage/microglia participating in demyelination-remyelination in this acute demyelination model. We found a robust increase in proliferation of CD68+ cells in mutant compared to control mice at eight dpi (166.4 ± 24.3 vs. 79.3 ± 20.6 CD68+/Ki67+ cells per mm$^2$ in mutant and control mice, respectively, p=0.026) (*Figure 6A–C*). Upon CNS insult, microglia/macrophages are quickly activated, changing their shape from ramified to rhomboid. Rhomboid versus ramified polarization of total or activated microglia/macrophages was examined using respectively Iba1 (*Figure 6D–E*) and CD68 (*Figure 6F–H*) immunostaining. We observed a switch of the microglia/macrophage polarization among the whole Iba1 and CD68 population in favor of the rhomboid phenotype in mutant mice compared to control at eight dpi. This effect was quantified for activated microglia (ratio of rhomboid/ramified CD68+ cells of 0.28 ± 0.05 in control vs. 0.66 ± 0.1 in mutant mice, p=0.038) (*Figure 6H*). While the activation phenotype tended to decrease between 4 and 8 dpi in control mice, it tended to increase in mutant animals. We then evaluated the expression level of Cox2, a marker of pro-inflammatory (M1) microglia/macrophage (*Chhor et al., 2017*) and observed a significant 77% increase in the number of Cox2+ microglial cells in mutant mice compared to control (p=0.01), indicating a delay in the pro-inflammatory (M1) to pro-regenerative (M2) switch in the absence of Ndst1 in oligodendroglia (*Figure 6I–K*). These results demonstrate that *Ndst1* deletion in the Olig2 population is sufficient to enhance microglia/macrophage proliferation and activation at the lesion site at the onset of remyelination.

## Shh binds to HS around the focal LPC-induced demyelinated lesion in the corpus callosum

As previously mentioned, HSPGs form a scaffold that shapes the distribution and activity of numerous growth factors and morphogens during development and provide environmental support for regenerative responses following CNS injury. Among several known HSPG-binding morphogens, Shh was previously identified as a positive regulator for myelin repair (*Ferent et al., 2013*; *Zakaria et al., 2019*). In order to determine whether lesion-induced HS enrichment around the lesion site could influence Shh distribution (hence signaling), we used an Alkaline Phosphatase (AP) tagged version of Shh (AP-SHH) to directly assay its binding capacity in demyelinating context. Knowing that the CW sequence serves as a major HS-binding site for Shh (*Carrasco et al., 2005*; *Rubin et al., 2002*), we also used AP-SHH recombinant proteins deleted for the CW sequence (AP-SHH-CWdeleted), as a control. Probes were incubated on fresh brain sections obtained 4 days after LPC injections. AP-Shh binding was observed in the cortex in healthy conditions (data not shown) and after

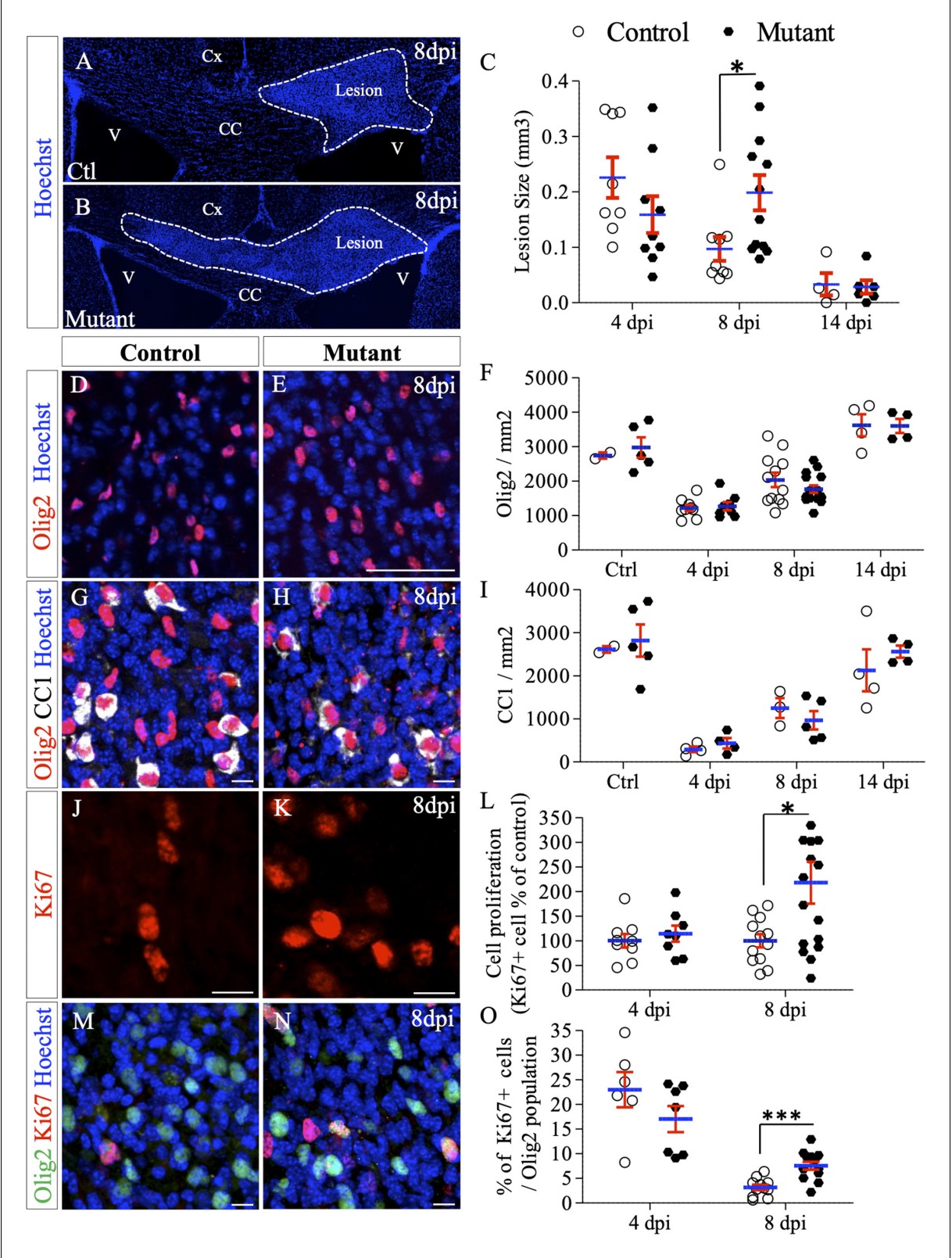

**Figure 5.** Deletion of *Ndst1* in Olig2+ cells affects lesion size and OPC mobilization after LPC-induced demyelination of the corpus callosum. (A–B) Representative images of the lesion site (delineated by white dashed lines) in the corpus callosum of control (A) and mutant (B) mice at eight dpi illustrating the enlargement of the lesion size in mutant mice compared to control mice. (C) Quantitative analysis of the lesion size at 4, 8 and 14 dpi (n = 8,9,4 control and n = 9,12,6 mutant mice respectively). (D–E) Oligodendroglia labeled by Olig2 staining within the demyelinated area at eight dpi

*Figure 5 continued on next page*

Figure 5 continued

(E) compared to control mice (D). (F) Olig2 mean cell density in healthy (CTL) or demyelinated control and mutant mice at 4, 8, 14 dpi. (G–H) Mature OLG co-labeled by Olig2/CC1 within the demyelinated lesion at eight dpi in control (G) and mutant (H) mice. (I) Quantification of mean cell density of Olig2+/CC1+ cells within the demyelination lesion in healthy (CTL) or demyelinated control and mutant mice at 4, 8, 14 dpi. (J–K) Ki67+ immunolabeling shows the proliferation status of cells within the lesion 8dpi in control (J) and mutant (K) mouse. (L) Graph represents the cell proliferation (Ki67+ cells) in mutant relative to control mice at 4 and 8 dpi (n = 9,12 control and n = 8,16 mutant mice respectively). (M–N) Co-immunolabelling of Olig2 and Ki67 showing OPC proliferation in control (M) and mutant (N) mouse 8dpi. (I) Quantification of proliferating OPC (Ki67+/olig2+ cells) in lesion sites at 4 and 8 dpi (n = 6,11 control and n = 7,13 mutant mice respectively). Error bars represent S.E.M. *p<0.05, ***p<0.001, non-parametric Mann-Whitney test (independent two group comparisons). Scale bars: 50 µm in A, B, D, E and 10 µm in, G, H, J, K, M and N. Source files of quantitative analyses are available in the *Figure 4—source data 1*.

The online version of this article includes the following source data for figure 5:

**Source data 1.** Source data for graphs in panels C, F, I, L, and O.

lesion (*Figure 7B,B'*). While no AP-Shh binding was observed in healthy or uninjured contralateral corpus callosum, AP-Shh binding delimited a clear belt surrounding the lesion site in the corpus callosum after LPC injection (*Figure 7B–B'*). In contrast, AP-Shh-CWdeleted did not bind around the same lesion on adjacent sections (*Figure 7C–D*). As AP-Shh binding depends on the integrity of its HS-binding motif, this indicates that endogenous Shh localization and concentration may be controlled by HS production by peri-lesional OLG.

In order to assess whether SHH signaling was indeed reduced in *Ndst1* mutant mice compared to control mice, we quantified *Ptch1* (the main SHH receptor) expression around LPC-induced demyelination lesions at eight dpi using RNAscope (*Figure 7E–G*). Ndst1 mutant mice exhibited 38% decrease in Ptch1 expression compared to control mice (8.6 ± 0.7 vs 13.8 ± 3.0 dots/cells in mutant and control mice respectively, n = 5 mice/group) although it did not quite reach significance (p=0.07) (*Figure 7E*). Altogether these results suggest that lack of NDST1 in OLG lineage attenuates SHH signaling following demyelination insult.

## NDST1 is expressed by oligodendroglia in multiple sclerosis lesions and correlates with lesion size and remyelination

To examine the relevance of our findings for multiple sclerosis (MS) physiopathology, we examined Ndst1 expression in MS tissue. We first probed the snRNAseq data provided by Jakel et al work (*Jäkel et al., 2019*), and observed that few cells express Ndst1 in both control and MS tissues but when the oligodendrocytes that express Ndst1 are compared, there is a trend to increased expression in MS tissue (*Figure 8—figure supplement 1*). Because such approach identifies around 15% only of nuclear RNA, we then directly examined the expression pattern of NDST1 protein in MS patient brain sections. Normal appearing white matter (WM), remyelinating, active, chronic active or chronic inactive lesions were analyzed. While NDST1 staining was very weak in control WM (without MS), we observed a significant increase of NDST1 labeling in MS patients WM (*Figure 8A–B*). Comparison of healthy control, MS normal appearing WM and MS lesions showed that there is a significant increase of NDST1 staining in multiple sclerosis lesions vs. control (p=0.0014) (*Figure 8C*). Comparison of each MS lesion with its surrounding normal appearing WM using a paired t test, revealed that there is significantly more NDST1 labeling in MS lesions compared to their surrounding normal appearing WM (paired two-tailed t test, $t_9$ = 3.39, p=0.0095). However, NDST1+ cells were distributed evenly throughout the lesion, rather than forming a delimiting band (*Figure 8—figure supplement 2D*).

We then performed double-labeling immunohistochemistry to characterize NDST1-positive cells in various types of lesions and normal appearing WM, using OLIG2 for oligodendroglia (*Figure 8D*), GFAP for astrocytes (*Figure 8E*), NEUN for neurons (*Figure 8F*) and IBA1 for microglia/macrophages (*Figure 8G*). Quantitative analysis showed that the majority of NDST1 cells were oligodendroglia in all types of lesions (remyelinated, active, chronic active, chronic inactive) and in normal appearing WM (*Figure 8H*).

The vast majority of OLIG2+ cells in the lesions expressed NDST1, with a gradual reduction in the proportion of OLIG2+NDST1+ cells as lesions become more chronic, and with significantly fewer in control tissue (*Figure 8I*; p=0.016). The number of NDST1+ oligodendroglia in each lesion was inversely correlated with the lesion's size (*Figure 8J*). As blocks of MS tissue contained multiple

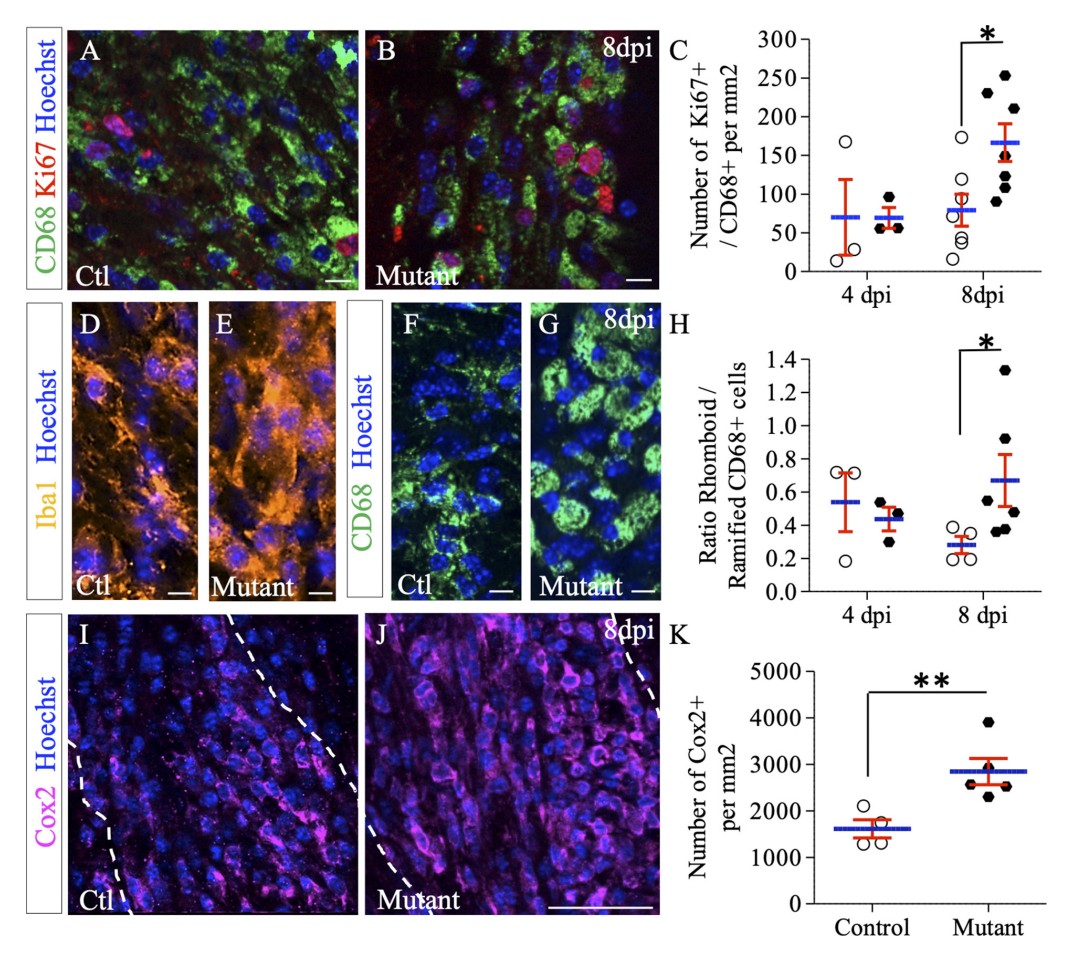

**Figure 6.** Effect of *Ndst1* deletion on microglia/macrophage activation. (**A–B**) CD68+/Ki67+ co-immunolabeling shows the proliferation status of activated microglia/macrophages. (**C**) Quantification of proliferating microglia/macrophages (Ki67/CD68+ cells) in lesion sites at 4 and 8 dpi (n = 3,7 control and n = 3,7 mutant mice respectively). Iba1 (**D–E**) and CD68 immunolabeling (**F–G**) shows the increase in rhomboid-polarized microglia/macrophages in the demyelinated area of mutant mice at eight dpi. (**H**) Quantification of the ratio of rhomboid/branched CD68+ cells in lesion sites at 4 and 8 dpi (n = 3,4 control and n = 3,6 mutant mice respectively) showing a switch of the microglia/macrophage polarization in favor of the rhomboid phenotype in mutant mice at eight dpi. (**I–J**) Cox2 immunolabeling shows an increase in this M1 phenotype marker at 8dpi in mutant mice. (**K**) Quantification of Cox2+ cells in lesion sites at 8dpi (n = 4 control and n = 5 mutant mice). Error bars represent S.E.M. *p≤0.05, non-parametric Mann-Whitney test (independent two group comparisons). Scale bars: 50 µm in I-J and 10 µm in A-B, D-E, F-G. Source file of quantitative analyses are available in the *Figure 5—source data 1*.

The online version of this article includes the following source data for figure 6:

**Source data 1.** Source data for graphs in panels C, H and K.

lesions and sometimes we had multiple blocks from the same patient (see *Table 1*), we gave each patient an overall remyelination ability score corresponding to how many lesions in the blocks from that patient were remyelinated, or likely to remyelinate if the patient had survived. Here, we were aiming to see whether patients considered being 'good remyelinators' using this score express more NDST1. A lesion was given a score of 3 points (complete remyelination), two points (active - likely to remyelinate), one point (chronic active less likely to remyelinate) and 0 points (chronic inactive - unlikely to remyelinate). The total score of all the lesions per patient was then divided by the number of lesions per patient, to allow comparisons. We showed that NDST1+ cell density positively correlated with patient's score of remyelination ability (*Figure 8K*). These data reveal that MS tissues with a higher repair potential (containing most active and remyelinated lesions) display a high number of NDST1+ cells therefore suggesting that higher numbers of NDST1+ cells in a lesion may provide a positive environmental support for myelin repair.

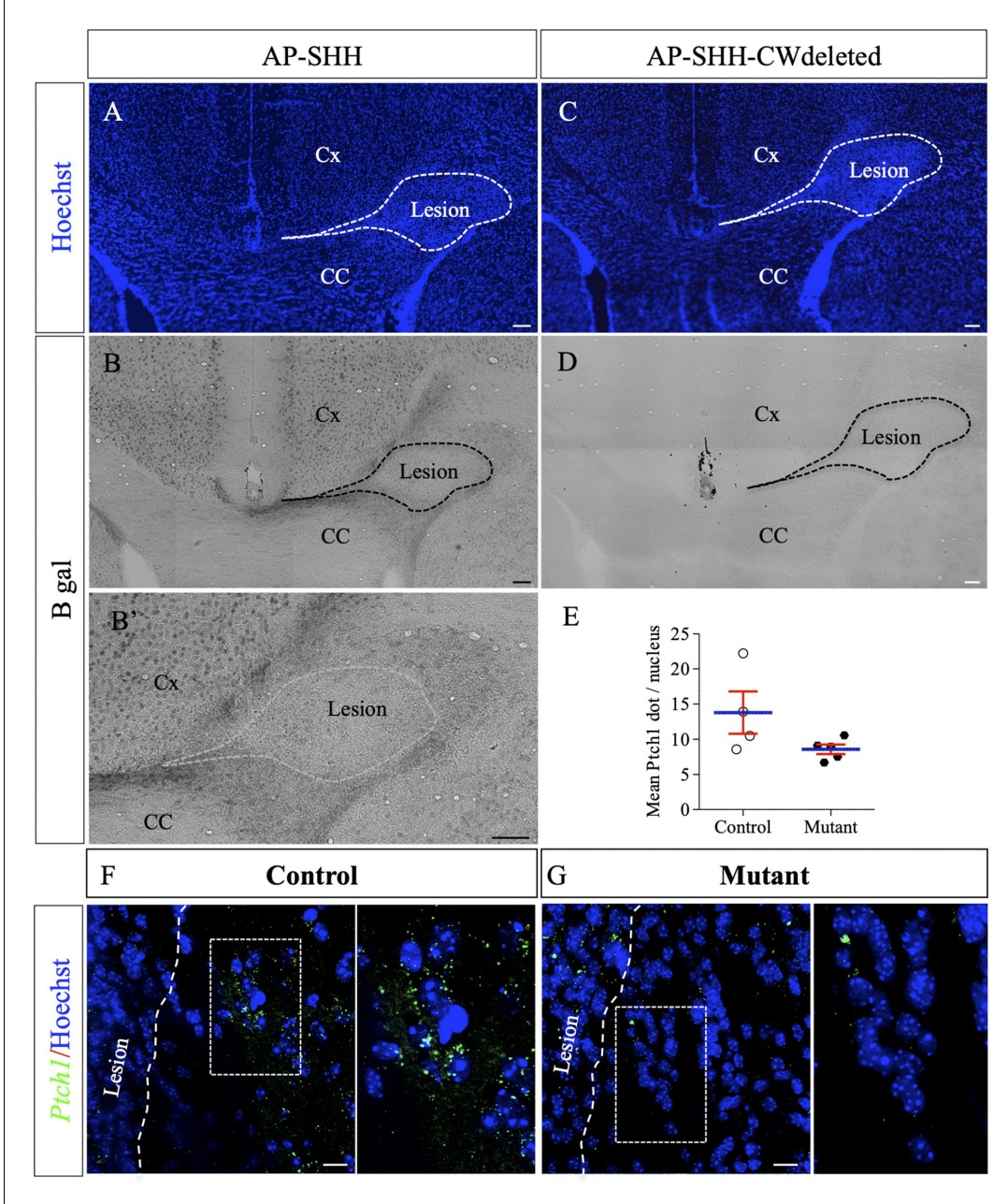

**Figure 7.** AP-tagged Shh protein binds to HS concentrated around LPC-induced lesions in the corpus callosum. Representative images of adjacent serial coronal sections derived from control mice 4 days after LPC injection and incubated with the fusion proteins AP-Shh-WT (**A–B'**) or AP-Shh-CW in which the CW sequence responsible for HS binding is absent (**C–D**) (n = 4). The lesion site is delineated by dashed lines. Staining using B-gal is clearly visible around the lesion after AP-Shh incubation (**B–B'**), while no staining is observed when the AP-Shh-CW deleted protein is used (**D**). These data show that Shh is concentrated around the lesion and that this distribution depends on the integrity of the HS binding motif. (**E**) Quantification of Ptch1 expression at 8dpi in control and mutant mice reported in number of dots per cell (n = 4 control and n = 5 mutant mice, p=0.07). (**F–G**) Illustration of Ptch1 expression in peri-lesional areas in control (**F**) and mutant (**G**) mice after labeling as detected by RNAscope technology. CC, corpus callosum; Cx, cortex. Scale bars: 100 μm in A-D. 10 μm in F and G. Source file of quantitative analysis is available in the *Figure 6—source data 1*.

The online version of this article includes the following source data for figure 7:

**Source data 1.** Source data for graph in panel E.

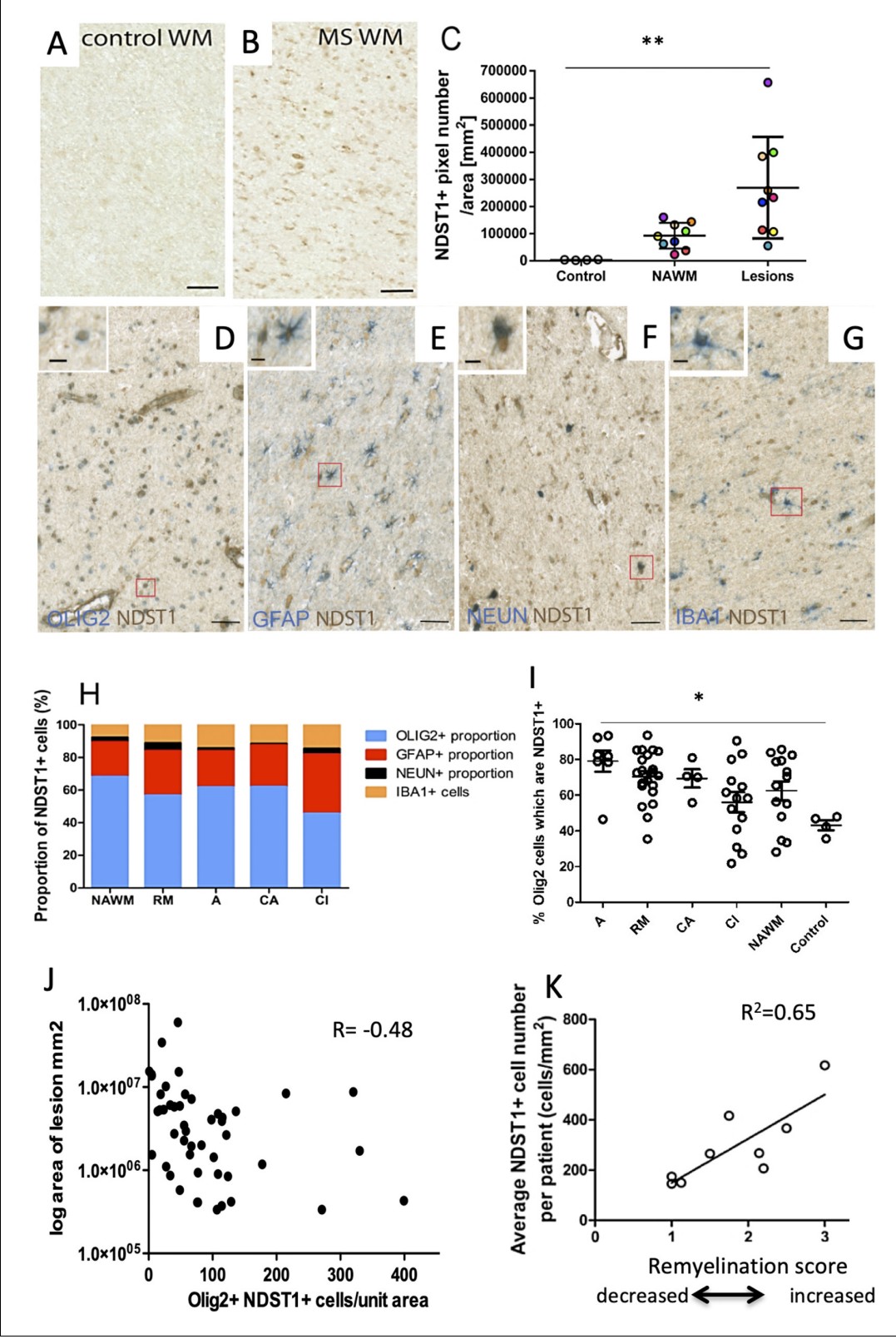

**Figure 8.** NDST1 is highly expressed in MS tissue and NDST1+OLIG2+ cell density negatively correlates with lesion size. (A–B) Representative images of NDST1 staining in control (A) and MS (B) WM. (C) Quantification of NDST1 labeling shows a significant over-expression of NDST1 in MS lesions (n = 9) compared to control tissue (n = 4) (Kruskal-Wallis test, H = 13.09, n = 4,9,9, p<0.01, means plus standard deviation). The colors represent

*Figure 8 continued on next page*

*Figure 8 continued*

paired samples from the same patients. (**D–G**) Representative images of immunostaining against NDST1 successively co-labeled with OLIG2+ for oligodendroglia (**D**), GFAP+ for astrocytes (**E**), NEUN+ for neurons (**F**), and IBA1+ for microglia/macrophages (**G**). (**H**) Quantification of the proportions of different NDST1+ cell types in normal appearing WM and various MS lesions shows that NDST1 expressing cells are mainly oligodendroglia. (**I**) The proportion of OLIG2+ cells which is NDST1+ is significantly increased in active lesions compared to control (Kruskal-Wallis test, H = 13.92, n = 7,21,4,14,14 p<0.05). Overall, the majority of OLIG2+ cells are NDST1+ in MS lesions and NAWM while this is not true in control brain tissue. (**J**) The number of oligodendroglia expressing NDST1 is inversely correlated to lesion size. (**K**) NDST1+ cell numbers positively correlate with the remyelination score assigned to each patient, summing all lesions within blocks from the same MS patients (see Materials and methods). NAWM, normal appearing white matter; RM, remyelinated lesion; A, active lesion; CA, chronic active lesion; CI, chronic inactive lesion. Scale bars represent 50 µm (**A–B**) or 10 µm (**D–G**). Source files of quantitative analyses are available in the *Figure 7—source data 1*.

The online version of this article includes the following source data and figure supplement(s) for figure 8:

**Source data 1.** Source data for graphs in panels C, I, J and K.
**Figure supplement 1.** Comparisons of Ndst1 expression levels in control and MS brain tissue from all nuclei (A), or just oligodendroglia (B) showing a tendency to increased levels in MS samples.
**Figure supplement 2.** NDST1 staining is specific and no lesion belt effect is observed in human brain.

## Discussion

In this study, we investigated the role of Ndst1 and HS after demyelination. Using LPC-induced demyelination of the corpus callosum in mouse, we showed for the first time that Ndst1-dependent N-sulfate sugar modifications occur at the onset of demyelination and during remyelination. These modifications limit the extension of demyelination and create a permissive substrate enhancing remyelination. First, we show that *Ndst1* is almost exclusively expressed by oligodendroglia present at the margin of the lesion delimiting the lesion from the intact corpus callosum. Second, we found that the conditional deletion of *Ndst1* in the Olig2 population concomitantly triggers an enlargement of the LPC-induced demyelinated area, alters OPC mobilization and favors the pro-inflammatory (M1) phenotype in microglia. We propose that these effects could be mediated through HS dependent binding of the morphogen Shh which has been shown to be a positive regulator of myelin repair through increased oligodendrogenesis and microglial regulation (*Ferent et al., 2013*; *Zakaria et al., 2019*). Finally, using MS brain tissue samples, we show that NDST1 is up-regulated, especially within lesions and that the density of NDST1+ cells in these lesions is negatively correlated

**Table 1.** Classification and characteristic of human post-mortem samples.

| | Patient | Sex | Age (years) | MS type | Disease duration (years) | Time to post mortem (h) | Number of lesions | Active | Chronic active | Chronic inactive | Remyeli-nating |
|---|---|---|---|---|---|---|---|---|---|---|---|
| MS | MS100 | M | 46 | SP | 8 | 7 | 6 | 0 | 0 | 4 | 2 |
| | MS121 | F | 49 | SP | 14 | 24 | 2 | 1 | 0 | 1 | 0 |
| | MS122 | M | 44 | SP | 10 | 16 | 2 | 1 | 1 | 0 | 0 |
| | MS136 | M | 40 | SP | 9 | 10 | 9 | 1 | 0 | 3 | 5 |
| | MS154 | F | 34 | SP | 21 | 12 | 4 | 2 | 0 | 1 | 1 |
| | MS176 | M | 37 | PP | 27 | 12 | 7 | 0 | 0 | 2 | 5 |
| | MS187 | F | 57 | SP | 27 | 13 | 4 | 0 | 0 | 0 | 4 |
| | MS207 | F | 46 | SP | 25 | 10 | 8 | 0 | 3 | 3 | 2 |
| | MS230 | F | 42 | SP | 19 | 31 | 4 | 2 | 0 | 0 | 2 |
| Control | CO14 | M | 64 | - | - | 26 | - | - | - | - | - |
| | CO25 | M | 35 | - | - | 22 | - | - | - | - | - |
| | CO28 | F | 60 | - | - | 13 | - | - | - | - | - |
| | CO39 | M | 82 | - | - | 21 | - | - | - | - | - |
| Total | | | | | | | 46 | 7 | 4 | 14 | 21 |

to lesion size, and positively correlated to the patient's potential remyelination ability. Our data suggest that Ndst1/HS expressed by oligodendrocytes around the lesion create a protective and permissive environment playing a positive role in myelin repair.

HS and chondroitin sulfates are the two main classes of sulfated proteoglycans constituting the extracellular space. Chondroitin sulfates are strongly expressed by astrocytes and microglia providing a hostile environment impeding regeneration and remyelination following brain injury (*Siebert and Osterhout, 2011*; *Pendleton et al., 2013*; *Deng et al., 2015*; *Lau et al., 2012*). Enzymatic degradation of chondroitin sulfate proteoglycans using chondroïtinase ABC treatment in vivo promotes OPC mobilization and remyelination (*Lau et al., 2012*). In vitro, chondroitin sulfates reduce OPC maturation (*Karus et al., 2016*) and in vivo the use of a CSPG synthesis inhibitor promotes OPC maturation and accelerates remyelination following focal demyelination in mice (*Keough et al., 2016*). A recent study has shown that Surfen, a proteoglycan binding agent, reduces inflammation and delays remyelination. (*Warford et al., 2018*). Our study provides evidence for a beneficial effect of HS proteoglycans (and upstream enzyme Ndst1) on limiting demyelinated lesion size and promoting remyelination. Here we propose that HS on mature OLG changes the amount/stability of soluble factors, and thus the recruitment and proliferation of surrounding cells (such as OPC or microglia). Thus, OLG modify their environment which in turn regulates the behavior of cells in the lesioned CC including microglia and OPC. Therefore, two classes of sulfated proteoglycans have opposite effects on myelin repair; one being detrimental (chondroitin sulfates), and the other (HS) having a beneficial effect. Interestingly, both are found at the margin of the lesion, but while the chondroitin sulfates are secreted by astrocytes and microglia, HS are expressed by oligodendroglia. Our data show that the majority of *Ndst1+* cells at the margin of the LPC-induced lesion are mature CC1+ OLG revealing that mature OLG around a demyelination lesion, respond to post lesional cues. Interesting, recent data have shown that pre-existing mature OLG can also participate to myelin regeneration in human and rodent by forming new myelin sheaths further indicating that mature OLG have an active role in myelin repair (*Yeung et al., 2019*; *Duncan et al., 2018*).

This OLG response to nearby demyelination appears propitious to regeneration by restricting the lesion spread, favoring OPC mobilization and modulating microglia response.

Macrophages/microglia participate to myelin repair through myelin debris clearance and the secretion of regenerative factors that altogether promote the recruitment of OPC, their proliferation and differentiation into mature myelin-forming cells. Activated microglia also produces various pro-inflammatory mediators (cytokines, chemokines...) that may affect OPC since they express a battery of receptors (*Peferoen et al., 2014*). Recent studies have shown that efficient remyelination required the dynamic regulation of functional microglia phenotype (*Miron et al., 2013*; *Lloyd et al., 2019*). Microglia respond to demyelination with initial pro-inflammatory phenotype (M1) followed later by pro-regenerative phenotype (M2) which actively contributes to myelin repair (*Miron et al., 2013*; *Olah et al., 2012*). Interestingly, this transition from pro-inflammatory to pro-regenerative phenotype is a rate-limiting step in the repair process since intra-lesional depletion of pro-regenerative microglia blocks oligodendrocyte differentiation and delays the regenerative process (*Miron et al., 2013*). In our study, lack of Ndst1/HS from Olig2+ cells in transgenic mice leads to increased microglial proliferation, a strongly activated rhomboid polarization of CD68-expressing cells and an increased density of pro-inflammatory (M1) microglia at an early stage of the remyelination phase (eight dpi). Altogether these effects may contribute to enlargement of the demyelinated lesion and delayed myelin repair. This non-cell autonomous effect observed on microglia activation could in turn disturb OPC mobilization and thus modify the repair process (*Miron et al., 2013*).

The beneficial action of HS may be related to their ability to bind numerous growth factors and morphogens, as observed during development (*Häcker et al., 2005*). HS can act as co-receptors for these ligands or are involved in the stabilization and/or local concentration of ligands in the extracellular space, which modulates cell signaling. Ndst1 global knock out mice display developmental defects that mainly resemble those found in embryos deficient for Shh or FGFs (*Pallerla et al., 2008*). FGF and Shh implication in myelin regeneration and glial reactivation have been extensively examined in mouse models of demyelination (for review, see *El Waly et al., 2014*). A role of both factors in remyelination was first inferred by correlating their spatial and temporal upregulation after demyelination particularly in LPC-induced demyelination (*Ferent et al., 2013*; *Gudi et al., 2011*; *Hinks and Franklin, 1999*; *Tourbah et al., 1992*).

Concerning FGF, conflicting results have been published on its activity (*Ruffini et al., 2001*; *Dehghan et al., 2012*; *Kumar et al., 2007*; *Armstrong et al., 2002*; *Zhou et al., 2012*; *Kang et al., 2019*). A recent report has addressed this issue using the simultaneous ablation of both FGFR1 and FGFR2 specifically in cells from the oligodendrocyte lineage (*Furusho et al., 2015*). This study revealed that FGF signaling is not required for myelin regeneration in acute models of demyelination including LPC-induced demyelination of the spinal cord and cuprizone intoxication (*Furusho et al., 2015*). Overall, in all these analyses the phenotypes observed after LPC-induced demyelination never recapitulate (even partially) the phenotype observed in the present report using *Olig2-Cre; Ndst1* *Flox/Flox* mice. This suggests that FGF is probably not the main ligand regulated by HS activity during myelin repair in this model.

By contrast, blocking Shh activity leads to an increase in demyelinated lesion size and an altered OPC mobilization after LPC-induced demyelination (*Ferent et al., 2013*) similar to what we observe in *Olig2-Cre; Ndst1* *Flox/Flox* mice. However, these effects persist at later time points at the end of remyelination, and OPC maturation is also inhibited after Shh inactivation (*Ferent et al., 2013*). Of note, recent in vitro findings show that the proteolytic processing of Shh required for signaling pathway activation (*Ohlig et al., 2012*) is finely regulated by HS chains (*Ortmann et al., 2015*), perhaps influencing Shh concentration and/or spreading. Here we show that Shh binds to HS around demyelinated lesions in mouse. Thus, we propose that HS removal may delay the local accumulation around the demyelinated lesion of Shh produced by OLG (*Ferent et al., 2013*) and/or delay Shh activation and spreading, leading to a reduction in signaling as suggested by reduced Ptch1 expression at early time-points. Later, the sustained production of Shh may overtake the absence of HS, leading to efficient recovery. Interestingly, *Ferent et al., 2013* have shown that the main Shh responding cells (cells expressing Gli1 and/or Smo) after LPC-induced demyelination of the corpus callosum are OLG and microglia. These observations further support the idea that sustained microglia and OPC activation in the absence of HS are at least in part due to altered Shh signaling.

Finally, we can confirm that NDST1 is also over-expressed in human MS brain samples, mostly in Olig2-positive oligodendroglia. High levels of HS proteoglycans and chondroitin sulfate proteoglycans have been previously associated with inflammatory CNS diseases such as MS (*Briani et al., 2002*; *Berezin et al., 2014*; *van Horssen et al., 2006*; *Satoh et al., 2009*). Here, we show that NDST1 is upregulated within and surrounding MS lesions in post mortem tissue compared to control. NDST1 expression is significantly higher in MS lesions compared to surrounding normal appearing white matter (NAWM), irrespective of lesion type (remyelinated, active, chronic active or chronic inactive). The distribution of the NDST+ cells was different from the mouse model, in that we saw no surrounding band of positive cells, but instead the entire lesions contained positive cells, though the majority of these cells were OLIG2+ oligodendroglia. This difference may be related to the timing of examination of the tissue after the lesion onset, which is later in the human tissue, and secondary to poorer repair in humans. In MS lesions, a large proportion of these OLIG2+ cells express NDST1, suggesting that at least some of these are mature, but we were unable to distinguish mature and immature oligodendroglia in this tissue (due to limitations in double-labelling using effective antibodies in human tissues). However, these observations still concur with our mouse data indicating that oligodendroglia respond to neighboring demyelination. Also consistent with our results showing increased lesion size in *Olig2-Cre+/-; Ndst1* *Flox/Flox* mice, in human samples we observed an inverse correlation between lesion size and density of NDST1-expressing Olig2+ cells. NDST1 cell density also positively correlates with a pathological score of potential remyelination ability in patients.

Overall, our results in mouse and human tissues suggest that NDST1/HS levels are an indicator of oligodendroglial reactivity after demyelination, and are involved in both limiting the size of the lesion and creating a permissive environment for myelin regeneration. Furthermore, this study shows for the first time that mature oligodendrocytes around lesion are active players during demyelination/remyelination by producing HS and thus modifying the local environment. This study will help improve understanding the neuropathology of MS in both limitation of damage and promotion of remyelination, which may in the future help target pharmacological approaches to potentiate myelin repair.

# Materials and methods

## Key resources table

| Reagent type (species) or resource | Designation | Source or reference | Identifiers | Additional information |
|---|---|---|---|---|
| Genetic reagent (*M. musculus*) | *Olig2$^{Cre}$* | PMID:18046410 | | B6D2F1J/Rj genetic background |
| Genetic reagent (*M. musculus*) | *Ndst1$^{flox/flox}$* | PMID:16020517 | | Dr. Kay Grobe (University of Münster, Münster, Germany) |
| Genetic reagent (*M. musculus*) | *Plp$^{gfp}$* | PMID:15906234 PMID:11756747 | | Dr. Bernard Zalc (University of Sorbonne, Paris, France) |
| Biological sample (*H. sapiens*) | Brain tissue from 9 MS patients | UK Multiple Sclerosis Tissue Bank (MREC/02/2/39) | | Postmortem unfixed frozen |
| Biological sample (*H. sapiens*) | Brain tissue from 4 Control patients | UK Multiple Sclerosis Tissue Bank (MREC/02/2/39) | | Postmortem unfixed frozen |
| Cell line (*H. sapiens*) | 293T HEK | ATCC | CRL3216 | |
| Transfected construct (*M. musculus*) | pWiz-AP-SHH | PMID:16020517 | | Production of AP-tagged SHH recombinant protein |
| Transfected construct (*M. musculus*) | pWiz-AP-SHH-CWdeleted | PMID:11959830 | | Production of AP-tagged deleted SHH recombinant protein |
| Antibody | Rabbit polyclonal anti-OLIG2 | Millipore | AB9610 | IF (1/1000) |
| Antibody | Rabbit polyclonal anti-OLIG2 | Sigma-Aldrich | HPA003254 | IF (1/100) |
| Antibody | Mouse monoclonal anti-APC (clone CC1) | Calbiochem | OP-80 | IF (1/400) |
| Antibody | Rat monoclonal anti-PDGFRa (clone APA5) | Millipore | CBL1366 | IF (1/250) |
| Antibody | Mouse monoclonal anti-MBP | Millipore | MAB384 | IF (1/500) |
| Antibody | Mouse monoclonal anti-Ki67 | BD Pharmingen | 556003 | IF (1/500) |
| Antibody | Rabbit polyclonal anti-Caspase 3 | Cell Signalling | 9661 | IF (1/200) |
| Antibody | Rabbit polyclonal anti-GFAP | Dako | Z0334 | IF (1/400) |
| Antibody | Goat polyclonal anti-IBA1 | Abcam | Ab5076 | IF (1/500) |
| Antibody | Rabbit polyclonal anti-IBA1 | Wako Chemicals | 019–19741 | IF (1/500) |
| Antibody | Rat monoclonal Anti-CD68 | Abcam | Ab53444 | IF (1/400) |
| Antibody | Rabbit polyclonal anti-COX2 | Abcam | Ab15191 | IF (1/400) |
| Antibody | Mouse monoclonal IgM anti-N-sulfated motifs on HS chains (clone10E4) | Amsbio | 370255–1 | IF (1/500) |
| Antibody | Mouse monoclonal anti-NDST1 | Abcam | ab55296 | IF (1/50) |
| Antibody | Rabbit polyclonal anti-NeuN | Abcam | Ab104225 | IF (1/500) |

*Continued on next page*

*Continued*

| Reagent type (species) or resource | Designation | Source or reference | Identifiers | Additional information |
|---|---|---|---|---|
| Sequence-based reagent | *Ndst1_F* | Eurofins Genomics | RT-qPCR primers | gctggacaagatcatcaatgg |
| Sequence-based reagent | *Ndst1_R* | Eurofins Genomics | RT-qPCR primers | acacagtacttctacgactatcc |
| Sequence-based reagent | *Gapdh_F* | Eurofins Genomics | RT-qPCR primers | gggttcctataaatacggactgc |
| Sequence-based reagent | *Gapdh_R* | Eurofins Genomics | RT-qPCR primers | ctggcactgcacaagaagat |
| Sequence-based reagent | *plp/dm20* | PMID:9373029 | | Probe for ISH |
| Sequence-based reagent | *Ndst1* | PMID:16020517 | | Probe for ISH |
| Sequence-based reagent | *Ptch1* | Advanced Cell Diagnostics | 402811-C2 | Probe for RNAScope |
| Peptide, Recombinant Protein | Human NDST1 | Abcam | ab116875 | |
| Commercial assay or kit | RNAscope Multiplex Fluorescent kit | Advanced Cell Diagnostics | 323133 | |
| Commercial assay or kit | DAB Peroxidase (HRP) Substrate Kit (with Nickel) | Vector Laboratories | SK-4100 | |
| Commercial assay or kit | VECTOR Blue AP Substrate Kit | Vector Laboratories | SK-5300 | |
| Commercial assay or kit | ImmPRESS-AP Anti-Rabbit IgG Polymer Detection Kit | Vector Laboratories | MP-5401 | |
| Commercial assay or kit | ImmPRESS HRP Anti-Mouse IgG Polymer Detection Kit | Vector Laboratories | MP-7402 | |
| Chemical compound, drug | Lysolecithin | Sigma-Aldrich-Merck | L1381 | |
| Chemical compound, drug | Heparinase | Amsbio | 100700 | |
| Chemical compound, drug | Lipofectamine 2000 | Invitrogen | 11668–030 | |
| Chemical compound, drug | Vector Bloxall | Vector Laboratories | SP-6000 | |
| Software, algorithm | ImageJ | https://imagej.nih.gov/ij/ | | |
| Software, algorithm | Zen two lite | Zeiss | | |
| Software, algorithm | GraphPad Prism | https://graphpad.com | | |

## Animals and treatments

All experimental and surgical protocols were performed following the guidelines established by the French Ministry of Agriculture (Animal Rights Division). The architecture and functioning rules of our animal house, as well as our experimental procedures have been approved by the ''Direction Départementale des Services Vétérinaires'' and the ethic committee (ID numbers F1305521 and 2016071112151400 for animal house and research project, respectively). Surgery and perfusions were performed under ketamine (100 mg/kg, MERIAL, Lyon, France))/xylazine (10 mg/kg, BAYER, Puteaux, France) anesthesia. C57BL/6 wild-type and transgenic mice were successively used to characterize post-lesional expression of *Ndst1* and HS after demyelination and to investigate the impact of conditional deletion of *Ndst1* in the Olig2-positive cell population. Heterozygous *Olig2-Cre+/-* (from B6D2F1J/Rj genetic background) (*Dessaud et al., 2007*) and double transgenic *Olig2-Cre+/-;*

*Ndst1* <sup>Flox/Flox</sup> mice (*Grobe, 2005*; *Dessaud et al., 2007*) will be referred below as control and mutant mice, respectively. Mice expressing GFP under the control of the proteolipid protein (plp, a protein largely present in myelin) promoter were used in some experiments to better observe demyelination lesions (called thereafter *plpGFP* mice). Animals were housed under standard conditions with enrichment and access to water and food ad libitum on a normal 12 hr light/dark cycle.

## Human postmortem samples

Postmortem unfixed frozen tissues were obtained from the UK Multiple Sclerosis Tissue Bank via a UK prospective donor scheme with full ethical approval (MREC/02/2/39). Luxol fast blue (LFB) (staining myelin; *Figure 8—figure supplement 2C*) and Oil Red O (staining lipids phagocytosed by macrophages) were performed to characterize and classify the lesion types (*Boyd et al., 2013*). Active lesions have indistinct borders on LFB and lipid-laden macrophages/microglia. Chronic active lesions have a ring of lipid-laden macrophages/microglia and a core with few immune cells. Chronic inactive lesions have a distinct border on LFB and few immune cells. Finally, shadow plaques, thought to represent remyelination, have less intense staining on LFB. This classification was done by two independent researchers for a previous publication (*Boyd et al., 2013*). In this study, we used active (n = 7), chronic active (n = 4), chronic inactive (n = 14) and remyelinated (shadow) MS plaques (n = 21) from 14 blocks of brain tissue from 9 MS patients and 4 blocks of brain tissue from four controls with no neurological disease (*Table 1*).

## Focal demyelination in the corpus callosum and tissue processing

Focal demyelination was performed by stereotactic injection of Lysolecithin (LPC) (SIGMA-ALDRICH, St Louis, USA) as described previously (*Cayre et al., 2013*; *Magalon et al., 2007*). The corpus callosum from healthy or demyelinated mice, from the ipsilateral and contralateral side to the LPC-induced lesion, were dissected 7 days post injection (dpi) from 1 mm thick coronal slices in cold Hank's Balanced Salt Solution (GIBCO by life technologie, Paisley, UK) and processed for RT-qPCR analysis (*Ndst1* primers : exon 6 (forward, 5'-gctggacaagatcatcaatgg-3') and exon 7 (reverse, 5'-acacagtacttctacgactatcc-3'); for *Gapdh*: exon1 (forward, 5'-gggttcctataaatacggactgc-3') and exon2 (reverse 5'-ctggcactgcacaagaagat-3'). Primers from EUROFINS GENOMICS, Ebersberg, GERMANY). For histological analysis, mice were anesthetized and perfused with ice-cold 4% paraformaldehyde (FISHER SCIENTIFIC, Loughborough Leics, UK) in PBS (GIBCO by life technologie, Paisley, UK). Brains were post-fixed overnight in 4% paraformaldehyde in PBS and cut on a vibratome (Leica) in 4 series of coronal sections (50 µm thick) for immunofluorescence, or cryopreserved and cut with cryostat (20 µm thick) for in situ hybridization.

## In situ hybridization and immunohistochemistry

In situ hybridization was performed using *plp/dm20* (*Peyron et al., 1997*) and *Ndst1* probes (*Grobe, 2005*) as described in *Virard et al., 2006*. RNAscope Multiplex Fluorescent kit (323133; Advanced Cell Diagnostics) was used to detect Ptch1 mouse mRNA. Briefly cryosections were baked for 30 min at 60°C and dehydrated, incubated for 30 min at 40°C with protease III before incubation with RNAScope probe Ptch1 (402811-C2) for 2 hr at 40°C. Immunohistochemistry was performed as described in *Cayre et al., 2013*. The following primary antibodies were used: rabbit anti-Olig2 (AB9610; 1/1000, Millipore, USA), mouse anti-APC (CC1) (1/400; Calbiochem, USA), and rat anti-PDGFRα (CBL1366; 1/250; Millipore, USA) for oligodendroglial lineage cells; anti-MBP (mouse, 1/500, Chemicon, Millipore S.A.) for myelin sheaths; mouse anti-Ki67 (556003; 1/500; BD Pharmingen) for proliferating cells; rabbit anti-caspase 3 (9661; 1/200; Cell Signaling) for apoptotic cells, rabbit anti-GFAP (1/400) for astrocytes; goat anti-Iba1 (1/500, Abcam) for microglia and macrophages; rat anti-CD68 (1/400, Abcam) for activated microglia and macrophages; rabbit anti-Cox2 (1/400, Abcam) for proinflammatory M1 microglia/macrophage; mouse IgM anti-N-sulfated motifs on HS chains (10E4 antibody, 1/500; Seikagaku, Japan). For Olig2 and Ki67 immunofluorescence, antigen unmasking was performed by 20 min incubation in boiling citrate buffer (10mM pH6). For N-sulfated motifs labeling, floating sections from PFA perfused-brain were incubated for 2 hr 30 at 37°C in buffer (100 mM Sodium Chloride, 1 mM Calcium Chloride, 50 mM Hepes 5 µg, BSA pH 7) with or without Heparinase (3.3 mU from Flavobacterium heparinum, Seikagaku Kogyo Co. # 100700, Japan) (*David et al., 1992*) before permeabilization. Secondary antibodies coupled to alexa 488, 555 and

647 (1/500, Invitrogen Molecular Probes) were applied for 2 hr 30 at RT in a humid chamber. Sections were counterstained with Hoechst 33342 (1/500, Sigma).

## AP-Shh recombinant protein binding test in mice

Plasmids containing sequences for AP-tagged N-terminal WT or deleted Shh were produced by PCR and ligated into pWIZ vector as described in *Grobe, 2005*; *Rubin et al., 2002*. Briefly, plasmids were transiently transfected into HEK cells using lipofectamine 2000 (Invitrogen). Transfection proceeded for 3 hr. Culture supernatants were collected after 60 hr and filtered through 0.45 µm filters (Corning Incorporated, Durham, USA). Hepes 10mM pH7 was added to increase stability. Shh concentration was then evaluated measuring AP activity in culture supernatants. Preparations from mock-transfected HEK cells were generated and used as vehicle controls. The AP-Shh binding test was performed as described in *Rubin et al., 2002*.

Fresh frozen brain sections were post-fixed with ice-cooled methanol for 8 min. After rinsing with phosphate-buffered saline containing 4 mM $MgCl_2$ and blocking with 1% Bovine Serum Albumin (SIGMA-ALDRICH, St Louis, MO, USA) 1 hr at RT, frozen adjacent sections from healthy or demyelinated C57BL/6 mice were incubated with 5 nM of two AP tagged versions of Shh: AP-Shh recombinant proteins carrying the N-terminal CW sequence (AP-SHH), the main HS-binding site for Shh (*Rubin et al., 2002*; *Carrasco et al., 2005*), or lacking this motif (AP-SHH-CWdeleted). Sections were then washed with PBS to dissociate any low affinity interaction and endogenous phosphatases were inactivated by heating at 65°C for 2 hr. AP was revealed by incubating overnight in NBT (100 mg/ml)/BCIP (50 mg/ml) in 100 mM Tris pH 9,5 with 100 mM NaCl and 50 mM $MgCl_2$.

## Immunohistochemistry on human post-mortem tissue

Tissue was fixed in 4% paraformaldehyde in PBS for 30 min. Endogenous peroxidase and AP activity was blocked by 10 min incubation with Vector Bloxall (Vector, SP-6000 VECTOR LABORATORIES, Burlingame, USA). Slides were blocked with ready-to-use 2.5% normal horse serum from Vector secondary antibody kits for at least 20 min. Primary antibodies were incubated overnight in antibody diluent (Spring Bioscience, ADS-125) at 4°C. Primary antibodies used: mouse anti-NDST1 (1/50; Abcam, ab55296), rabbit anti-NeuN (1/500; Abcam, ab104225), rabbit anti-IBA1 (1/500; Wako chemicals, 019–19741), rabbit anti-Olig2 (1/100; Sigma, HPA003254). HS staining with the mouse IgM anti-N-sulfated motifs on HS chains (10E4 antibody, Seikagaku, Japan) did not give any signal on human tissue. NDST1 intensity was evaluated after a short exposition (exactly 2 min). All other stainings were fully developed. To ensure antibody specificity, the NDST1 antibody was pre-absorbed with human NDST1 recombinant protein (Abcam, ab116875), and added to tissue sections, with no staining seen (*Figure 8—figure supplement 2A–B*).

Secondary antibodies were incubated at RT for 1 hr. Staining was developed with a DAB Peroxidase (HRP) Substrate Kit (with Nickel), 3,3'-diaminobenzidine (Vector, SK-4100) and a VECTOR Blue AP Substrate Kit (Vector, SK-5300) as per manufacturer's guidelines. Secondary antibodies used: ImmPRESS-AP Anti-Rabbit IgG Polymer Detection Kit (Vector, MP-5401) and ImmPRESS HRP Anti-Mouse IgG Polymer Detection Kit, made in Horse (Vector, MP-7402). PBS washes were performed between each treatment.

## Microscopy and quantification

For mouse tissue analysis, imaging was performed with the Apotome system (Zeiss). The demyelinated area and cell counts were evaluated using Zen software (Zeiss). Immunofluorescent or in situ hybridization positive cells were counted in every fourth section through the whole demyelinated lesion per mouse and averaged for each mouse. Cell counts are presented as the mean of at least three mice. For RNAscope ISH, each punctate dot signal was counted around lesion (by using ROI and analyze particule Fiji Plugins) and reported to total nuclei number. For Heparan sulfate labeling quantification, 10E4-mmunopositive area was analyses on five sections per mouse. A constant exposure time was applied to all sections and for image acquisition, and the area occupied by 10E4+ labeling was quantified using ImageJ software. Lesion size was quantified by measuring the area of high density of nuclei in every fourth section through the whole demyelinated lesion per mouse. In this analysis, high density of nuclei was correlated with myelin loss visualized by MBP staining or by loss of fluorescence in *plpGFP* mice (*Figure 1—figure supplement 2*).

For the human post-mortem tissue analysis, slides were imaged using a ZEISS Axio Scan.Z1 slide scanner. One researcher marked out the lesion sites and normal appearing white matter (WM) as areas of interest, while another counted single positive (NDST1+ cells) and double positive cells (NDST+ cells and other brain cell markers combined as above) in these areas of interest (ensuring blinding of counting).

Myelin content was evaluated by double blind scoring of images taken from Plp immunostaining on brain sections (three photos per section and three sections per brain). Score of 4 was attributed to maximum myelination down to 0 for absence of myelin. The mean score for the control group was considered as 100%.

## Gene expression profile of demyelinated versus healthy mouse progenitors

This protocol is fully described in *Cayre et al., 2013*. Briefly, OPC from eight mice induced for experimental autoimmune encephalomyelitis (EAE mice) at the peak of paralytic symptoms and from eight adult healthy mice as controls were purified using magnetic cell sorting (Miltenyi Biotec). This experiment was replicated in an independent similar experiment. cDNAs were prepared and used (250 ng) as template for Cy3 and Cy5, combined and hybridized to Agilent Whole Mouse Genome Oligo Microarrays 4 Å~44K. Agilent Feature Extraction Software (FES) determines feature intensities and ratios (including background subtraction and normalization), rejects outliers and calculates statistical confidences (P-values). We obtained a gene list with all normalized Cy5/Cy3 log10 ratios, Cy5/Cy3 fold changes, sequence description and P-values. Microarray data are available at GEO with accession number GSE47486.

## Statistical analysis

All the presented values in mice are means ± S.E.M unless otherwise stated. Data were statistically processed with the non-parametric Mann-Whitney test (independent two group comparisons). $p < 0.05$ was considered significant and $p < 0.01$ highly significant. All measurements and subsequent evaluations were performed blind to the experimental group to which the animals belonged. For the human post-mortem tissue analysis, a d'Agostino and Pearson omnibus normality test was used to test whether the data fit a normal distribution and a parametric test were done only if all compared data sets passed the normality test. The NDST1+ cells in control versus multiple sclerosis WM was compared using a two-tailed Mann Whitney U test. Multiple sclerosis lesions and their surrounding WM were compared using a paired two-tailed t test. The absolute numbers of NDST1+ cells and double positive NDST1+ OLIG2+ cells in individual lesions/normal appearing WM were compared by Kruskal-Wallis test. As MS tissue blocks contained more than one lesion, and we had several blocks from the same patients, we gave each patient an overall remyelination ability score corresponding to how many lesions in the blocks from that patient were remyelinated, or likely to remyelinate if the patient had survived. Remyelinated lesions received an arbitrary three points, active lesions two points, chronic active lesions one point and chronic inactive lesions 0 points. This was divided by the number of lesions counted for each patient, to allow comparisons.

## Acknowledgements

We are grateful to E Traiffort, F Helmbacher and C Bertet for critical reading of the manuscript.

---

## Additional information

### Funding

| Funder | Grant reference number | Author |
|---|---|---|
| Centre National de la Recherche Scientifique | | Pascale Durbec |
| Aix-Marseille Université | Graduate student fellowship | Pascale Durbec |
| Fondation pour la Recherche Médicale | DEQ20140329501 | Pascale Durbec |

---

| Agence Nationale de la Recherche | France-bioimaging/PICSL infrastructure ANR-10-INSB-04-01 | Pascale Durbec |
| --- | --- | --- |
| Agence Nationale de la Recherche | ANR-15-CE16-0014-01 | Pascale Durbec |
| AM*DEX NeuroMarseille Institute | AMX-19-IET-004 | Pascale Durbec |

The funders had no role in study design, data collection and interpretation, or the decision to submit the work for publication.

### Author contributions

Magali Macchi, Conceptualization, Formal analysis, Investigation, Methodology, Writing - original draft; Karine Magalon, Conceptualization, Formal analysis, Investigation, Methodology, Writing - review and editing; Céline Zimmer, Conceptualization, Formal analysis, Writing - original draft; Elitsa Peeva, Bilal El Waly, Béatrice Brousse, Sarah Jaekel, Formal analysis; Kay Grobe, Formal analysis, Validation, Writing - original draft; Friedemann Kiefer, Resources; Anna Williams, Formal analysis, Supervision, Investigation, Methodology, Writing - original draft, Writing - review and editing; Myriam Cayre, Conceptualization, Formal analysis, Supervision, Validation, Investigation, Methodology, Writing - original draft, Writing - review and editing; Pascale Durbec, Conceptualization, Supervision, Funding acquisition, Validation, Methodology, Project administration

### Author ORCIDs

Bilal El Waly (iD) http://orcid.org/0000-0003-2991-3754
Kay Grobe (iD) http://orcid.org/0000-0002-8385-5877
Pascale Durbec (iD) https://orcid.org/0000-0002-9660-1809

### Ethics

Human subjects: Human postmortem unfixed frozen tissues were obtained from the UK Multiple Sclerosis Tissue Bank via a UK prospective donor scheme with full ethical approval (MREC/02/2/39). Animal experimentation: All experimental and surgical protocols were performed following the guidelines established by the French Ministry of Agriculture (Animal Rights Division). The architecture and functioning rules of our animal house, as well as our experimental procedures have been approved by the 'Direction Départementale des Services Vétérinaires' and the ethic committee (ID numbers F1305521 and 2016071112151400 for animal house and research project.

### Decision letter and Author response

Decision letter https://doi.org/10.7554/eLife.51735.sa1
Author response https://doi.org/10.7554/eLife.51735.sa2

## Additional files

### Supplementary files
• Transparent reporting form

### Data availability
All data generated or analysed during this study are included in the manuscript and supporting files.

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
