## [Decision Letter]

**Acceptance summary:**

This manuscript explored the role of mature oligodendrocytes in regions surrounding demyelinating lesions in the production of heparin sulfates. These results have implications not only for limiting demyelination, but also in remyelination in MS and other demyelinating disorders.

**Decision letter after peer review:**

[Editors’ note: the authors submitted for reconsideration following the decision after peer review. What follows is the decision letter after the first round of review.]

Thank you for submitting your work entitled "Mature oligodendrocytes bordering lesions limit demyelination and favor myelin repair via heparan sulphate production" for consideration by *eLife*. Your article has been reviewed by three peer reviewers, one of whom is a member of our Board of Reviewing Editors, and the evaluation has been overseen by a Senior Editor. The following individual involved in review of your submission has agreed to reveal their identity: James Fawcett (Reviewer #3).

Our decision has been reached after consultation between the reviewers. Based on these discussions and the individual reviews below, we regret to inform you that your work will not be considered further for publication in *eLife*, at least not in its present scope. As you will see from the individual reports there was certainly interest in your findings of an oligodendroglial response to a demyelinating lesion. However, the reviewers felt that at present the experiments aimed at mechanistic insight resulted mostly in correlative data. Perhaps, using genetic mouse models would be a strategy to reach stronger conclusions on causation. These would be time-consuming additional experiments and we do not invite a revised manuscript at this point. Any new submission would be independently assessed, possibly with new reviewers.

Reviewer #1:

This paper is interesting because it provides to my knowledge the first report of a molecular response of intact oligodendrocytes close to a demyelinating lesion, the induction of an enzyme (Ndst1) that increases the expression of heparan sulfate and thus HS-modifed proteins. Based on the phenotype of OPC/OL-specific Ndst1 cKO mice the authors suggest that this oligodendroglial response limits the extent of demyelination and promotes remyelination, possibly by locally increasing sonic hedgehog signaling. The authors also show a similar upregulation of NDST1 within (not around) human MS lesions, where the expression level inversely correlates with lesion size.

Major points:

1) Since primary development and myelination are not perturbed in the brains of Ndst1 cKO mice, another possible explanation of the data is that enhanced activation of microglia (or their altered phenotype) affects remyelination in a non-cell autonomous way.

2) Thus, the Shh data are plausible but only correlations, and one should not leave the impression that reduced Shh binding was shown to cause the delay of remyelination.

3) It is unclear how "remyelination" in the human MS autopsy samples was assessed (pale blue staining could also be incomplete demyelination). In fact, this may explain the different pattern of Ndst1 expression in mouse and human demyelinating lesions.

4) Figure 3. It is unclear what the anti-PLP staining is actually labeling outside the lesion area (in Figure 3B), as PLP is still a myelin marker

5) Figure 1C-H: In-situ hybrization of Ndst1 mRNA in demyelinaing lesions: these observations are critical. but lack quantification (required standard). The alignment of the "lesion" with the "belt" is not obvious when using only dashed lines. Why is Ndst1 expression not shown also as an overview type image (similar to Figure 1—figure supplement 2)? Same is true for Figure 2, which also lacks quantitation. While the "estimated" level of upregulation is stronger than in Ndst1 in Figure 1B, the IHC images are (at present) only case reports.

Reviewer #2:

This paper examines the role of heparan sulphate in remyelination by examining the expression of a key enzyme NDST1 and the consequences of an NDST1 conditional knockout in a mouse model of remyelination. In addition, the authors examine tissue from patients with MS. They conclude that HS containing proteoglycans are involved in remyelination as the enzyme is expressed around demyelinating lesions and its loss delays remyelination as evidenced by an increase in lesion size. Moreover there is a correlation between the degree of expression and the degree of remyelination in MS tissue.

While I think that the role of proteoglycans in remyelination is interesting, this paper is predominantly descriptive and correlative and doesn't provide sufficient mechanistic detail. The evidence for SHH binding in the region of putative HS expression is noteworthy but it is purely correlative. Equally the increase in microglial proliferation is interesting but not explored further.

My view would be that experiments are required to define one or more mechanisms by which HS is involved in remyelination, with gain and/or loss of function experiments clearly proving that the proposed mechanism is both linked to HS expression and regulates oligodendrocyte behaviour during remyelination in vivo.

Additionally I think there are three other areas that require attention.

First, the authors argue that the distribution around the experimental lesions is significant and that they have uncovered a novel function for surviving oligodendrocytes around lesions. It's notable however that they do not see this distribution in the MS tissue and I wonder whether their results simply reflect the lack of any surviving oligodendrocytes following LPC injection. In other words the distribution they see is simply a consequence of the model rather than having any great biological significance

Second, the characterization of the OPC response in the presence or absence of HS is very poor and studies of proliferation and differentiation, including the quantification of newly formed oligodendrocytes, is required

Third, a better characterization of the microglial phenotype (e.g. M1/M2 or similar) is required if the significance of this response is to be understood

Reviewer #3:

This is a thorough and interesting account of the role of HSPGs in remyelination. Many mechanisms of remyelination are known, but the idea that a particular sulfation change in HSPGs could be important is new. There is also data from human MS pathology in the paper, which gives clinical relevance.

1) The paper focuses on NDST which gives a particular form of sulfation to HS. There needs to be some information about the sulfation motif that is produced and any known information on relevant molecules that might bind to this form of HS. Shh is binding is shown, but it is unlikely to be the only effector of the local HS change.

2) If there is an increase in NDST, do the authors have information about any HS core proteins that might also be upregulated?

3) Why would a sulfation change in mature oligos increase myelination? Surely OPCs would be more relevant?

4) The authors show that there is some increase in Shh binding around lesions, but not that it does anything. The statement in the Discussion needs to be modified. Also they do not show local upregulation. Any published information that might link Shh or Shh inhibitor to remyelination needs to be stated. This part of the paper needs attention.

5) Do the authors believe that the changes they see in NDST etc. are specific to demyelination, or are similar changes seen with traumatic lesions and inflammation?

6) Figure 4: Is the increase in cell proliferation in the knockouts entirely accounted for by increased division of microglia?

7) Figure 6 shows some increase in Shh binding around the lesion, but also widespread binding elsewhere. Some comment is warranted.

[Editors’ note: further revisions were suggested prior to acceptance, as described below.]

Thank you for submitting your article "Mature oligodendrocytes bordering lesions limit demyelination and favor myelin repair via heparan sulphate production" for consideration by *eLife*. Your article has been reviewed by three peer reviewers, one of whom is a member of our Board of Reviewing Editors, and the evaluation has been overseen by Gary Westbrook as the Senior Editor. The following individual involved in review of your submission has agreed to reveal their identity: James W Fawcett (Reviewer #2).

The Reviewing Editor has drafted this decision to help you prepare a revised submission.

Summary

The authors have submitted a comprehensive revision of the manuscript, with several pieces of new data. These changes address the main issues raised by the reviewers, and the paper is considerably improved. However, the authors have not addressed the mechanistic underpinning of the proposed mechanism – i.e. the belt of HS synthesis and deposition in response to Ndst1 upregulation in oligodendrocytes around the lesion, and the proposal that binding of Shh promotes both, OPC proliferation and microglia polarization. Three points require further revisions.

1) We think a fairly simple IHC experiment with existing materials should provide direct mechanistic insight:

a) by showing that HS production around the LPC lesions is lost or reduced in the Ndst1 KO

b) by showing that after LPC lesion the Shh downstream targets in OPCs (and in other cells) are less activated in Ndst1 KO mice

c) by showing that M2 microglia are indeed increased in number.

2) The authors are also encouraged to check for Ndst1 upregulation in the available databases on single nuclei gene expression in MS tissue.

3) Finally, the role of microglia has been experimentally explored by additional experiments, but the possibility of non-cell autonomous effects on remyelination (by microglia) has not been adequately discussed, despite some reassurance in the rebuttal letter.

---

## [Author Response]

[Editors’ note: the authors resubmitted a revised version of the paper for consideration. What follows is the authors’ response to the first round of review.]

Reviewer #1:This paper is interesting because it provides to my knowledge the first report of a molecular response of intact oligodendrocytes close to a demyelinating lesion, the induction of an enzyme (Ndst1) that increases the expression of heparan sulfate and thus HS-modifed proteins. Based on the phenotype of OPC/OL-specific Ndst1 cKO mice the authors suggest that this oligodendroglial response limits the extent of demyelination and promotes remyelination, possibly by locally increasing sonic hedgehog signaling. The authors also show a similar upregulation of NDST1 within (not around) human MS lesions, where the expression level inversely correlates with lesion size.Major points:1) Since primary development and myelination are not perturbed in the brains of Ndst1 cKO mice, another possible explanation of the data is that enhanced activation of microglia (or their altered phenotype) affects remyelination in a non-cell autonomous way.

We fully agree with this interpretation of our results. As requested by reviewers, we have further clarified this point and better described microglia activation. We have performed new experiments to further characterize the proliferation and activation states of the macrophage/microglia participating in demyelination-remyelination in the acute demyelination model (See new version of Figure 5). We have shown:

1) An increase in proliferation of CD68+ cells in mutant compared to control mice at 8 dpi (Figure 5A-C).

2) A switch of the microglia/macrophage polarization among the whole Iba1 and CD68 population in favor of the rhomboid phenotype in mutant mice compared to control at 8 dpi (Figure 5H).

3) A significant increase in Cox2 expression (a marker of pro-inflammatory (M1) microglia/macrophage) in mutant mice compared to control, indicating a delay in the pro-inflammatory (M1) to pro-regenerative (M2) switch in the absence of Ndst1 in oligodendroglia.

These results demonstrate that *Ndst1* deletion in the Olig2 population is sufficient to enhance microglia/macrophage proliferation and activation at the lesion site at the onset of remyelination. The implication of Microglia (in a non cell autonomous effect) is presented in the Discussion.

2) Thus, the Shh data are plausible but only correlations, and one should not leave the impression that reduced Shh binding was shown to cause the delay of remyelination.

Previous reports from our lab and others have shown that Shh is a strong candidate for myelin repair acting both on oligodendrogenesis and regulation of microglia activation (Ferent et al., 2013; Zakaria et al., 2019). Indeed, Shh is produced by OLG and OPC at the onset of demyelination in lysophosphatidyl choline (LPC)-induced lesions. In this context, blocking Shh activity induces an increase in lesion size and a block in OPC proliferation and differentiation, and conversely Shh overexpression leads to the attenuation of the lesion extent and promotes oligodendrogenesis. Microglia express Shh receptors (Ferent et al., 2013) and Shh treatment reduces the number of activated microglia in LPC model. Thus, Shh production by OLG lineage cells could act both directly on OPC (cell autonomous) and on microglia activation (non cell autonomous). Since Shh is bound and enriched at the border of the lesion in a HS dependent manner, it is likely that this morphogen contributes to the role of Ndst1/HS in the regeneration process.

We made the mistake of not presenting these results (Ferent et al….) in the Introduction of the paper but only in the Discussion. To better underline these key findings concerning Shh for the understanding of our work, we described these results and references in the Introduction of new version of the manuscript.

A second possible candidate could be FGF since HS can also modulate its signaling and since Ndst1 global knock out mice display developmental defects that mainly resemble those found in embryos deficient for Shh or FGFs. Concerning FGF recent work using the simultaneous ablation of both *FGFR1* and *FGFR2* specifically in OLG revealed that FGF signaling is not required for myelin regeneration. Overall, in all the analyses modulating FGF signaling in acute model of demyelination the phenotypes observed never recapitulate the phenotype observed in the present report using Olig2-Cre; Ndst1 ^Flox/Flox^ mice. This suggests that FGF is probably not the main ligand regulated by HS activity during myelin repair in this model. These data from the literature were absent from the first version of the manuscript and are now added in the Discussion.

3) It is unclear how "remyelination" in the human MS autopsy samples was assessed (pale blue staining could also be incomplete demyelination). In fact, this may explain the different pattern of Ndst1 expression in mouse and human demyelinating lesions.

“Shadow” plaques where there are patches of pale blue staining on luxol fast blue, suggesting thinner myelin sheaths, have long been suggested to be remyelinated lesions. This assumption is based on their similarity to remyelinated lesions seen in animal models. These lesions are not surrounded by many myeloid cells, and these do not contain myelin debris, again suggesting that these are at least not recently demyelinated. Thus, we believe these are repaired lesions but will call these lesions shadow plaques, to cover all possibilities including remyelination or partial demyelination.

4) Figure 3. It is unclear what the anti-PLP staining is actually labeling outside the lesion area (in Figure 3B), as PLP is still a myelin marker

There is a misunderstanding. In Figure 3 for plp labeling a double in situ hybridization staining was performed. We make it clearer in the text by writing “We found that *Ndst1* expressing cells are immuno-positive for Olig2 (Figure 3A) and *Plp*+ using double in situ hybridization (Figure 3B)”.

5) Figure 1 C-H: In-situ hybrization of Ndst1 mRNA in demyelinaing lesions: these observations are critical. but lack quantification (required standard). The alignment of the "lesion" with the "belt" is not obvious when using only dashed lines. Why is Ndst1 expression not shown also as an overview type image (similar to Figure 1—figure supplement 2)? Same is true for Figure 2, which also lacks quantitation. While the "estimated" level of upregulation is stronger than in Ndst1 in Figure 1B, the IHC images are (at present) only case reports.

Concerning quantitative analysis. At this stage of the work (Figure 1 and 2) we describe the expression pattern of ndst1 and HS in the lesioned brain. Both are virtually absent in the healthy corpus callosum as well as in the contralateral part after lesion using ISH and IF. We thus decided to perform quantitative analysis using RT-qPCR, to compare ndst1 expression in the corpus callosum of healthy or demyelinated mice using 5 mice in each condition. These quantitative data are presented in Figure 1. In a second step, we characterized the phenotype of cells expressing ndst1 and performed quantitative analysis showing that 100% of ndst1 positive cells are Olig2 positive.

As mentioned in the manuscript Ndst1 expression is monitored by in situ hybridization. This technic is not compatible with good quality myelin immunostaining. Thus, in order to delimit precisely lesions in the LPC model, we use Hoechst staining since the lesion is also characterized by an increase in cell density due to glia and microglia proliferation. In order to validate this approach we have performed injection of LPC in a reporter mouse (plp-GFP) and show that the limit of the lesion visualized by loss of GFP fluorescence strictly corresponds with the area exhibiting strong increase in cell density. These data are presented in Figure 1—figure supplement 2. Thus the alignment of the lesion was based on cell density after in situ hybridization.

Reviewer #2:This paper examines the role of heparan sulphate in remyelination by examining the expression of a key enzyme NDST1 and the consequences of an NDST1 conditional knockout in a mouse model of remyelination. In addition, the authors examine tissue from patients with MS. They conclude that HS containing proteoglycans are involved in remyelination as the enzyme is expressed around demyelinating lesions and its loss delays remyelination as evidenced by an increase in lesion size. Moreover there is a correlation between the degree of expression and the degree of remyelination in MS tissue.While I think that the role of proteoglycans in remyelination is interesting, this paper is predominantly descriptive and correlative and doesn't provide sufficient mechanistic detail. The evidence for SHH binding in the region of putative HS expression is noteworthy but it is purely correlative. Equally the increase in microglial proliferation is interesting but not explored further.My view would be that experiments are required to define one or more mechanisms by which HS is involved in remyelination, with gain and/or loss of function experiments clearly proving that the proposed mechanism is both linked to HS expression and regulates oligodendrocyte behaviour during remyelination in vivo.Additionally I think there are three other areas that require attention.First, the authors argue that the distribution around the experimental lesions is significant and that they have uncovered a novel function for surviving oligodendrocytes around lesions. It's notable however that they do not see this distribution in the MS tissue and I wonder whether their results simply reflect the lack of any surviving oligodendrocytes following LPC injection. In other words the distribution they see is simply a consequence of the model rather than having any great biological significance

First, as shown in Figure 1—figure supplement 1, we have observed the same pattern of Ndst1 expression in the EAE model were *Ndst1* is up-regulated by the Olig2+ cell population in close proximity to inflammation sites in corpus callosum. Since we have observed this expression in both mouse models (LPC and EAE), it is probably relevant.

Concerning the distribution of Ndst1 expression in the LPC model: Even though OLG are massively killed within the lesion, remaining OLG in close proximity to the lesion do express ndst1 while OLG remote from the lesion in the CC do not. This shape “as a belt” is may be linked to the model but the presence of reactive mature OLG ndst1+ at the lesion site observed in all models including human is biologically relevant.

This point has been discussed in the new version of the manuscript.

Second, the characterization of the OPC response in the presence or absence of HS is very poor and studies of proliferation and differentiation, including the quantification of newly formed oligodendrocytes, is required

We have performed new experiments presented now in Figure 4 in the new version of the manuscript. We performed a quantification of proliferating of OPC (Ki67+/olig2+ cells) in lesion sites at 4 and 8 dpi. We show that the dynamic of OPC proliferation is altered in mutant mice. Overall, Our data show that *Ndst1* expression in the Olig2+ population has no effect on initial demyelination (equivalent lesion size at 4 dpi) but protects the lesion from enlarging and participates in the control of OPC mobilization. We have included these data in the new version of the manuscript.

Third, a better characterization of the microglial phenotype (e.g. M1/M2 or similar) is required if the significance of this response is to be understood

See response to reviewer 1. We have performed new data showing that *Ndst1* deletion in the Olig2 population is sufficient to enhance microglia/macrophage proliferation and activation at the lesion site at the onset of remyelination. We demonstrate accentuated M1 phenotype in mutant mice compared to wild-type mice.

Reviewer #3:This is a thorough and interesting account of the role of HSPGs in remyelination. Many mechanisms of remyelination are known, but the idea that a particular sulfation change in HSPGs could be important is new. There is also data from human MS pathology in the paper, which gives clinical relevance.1) The paper focuses on NDST which gives a particular form of sulfation to HS. There needs to be some information about the sulfation motif that is produced and any known information on relevant molecules that might bind to this form of HS. Shh is binding is shown, but it is unlikely to be the only effector of the local HS change.

We have now included in the new version of the manuscript a full discussion concerning which factors could bind to HS in this context.

Concerning Shh binding, we have used an AP tagged version of Shh (AP-SHH) to directly assay its binding capacity in demyelinating context. We believe that an important piece of information is the data presented using the AP-SHH recombinant proteins deleted for the CW sequence (AP-SHH-CW deleted) since CW sequence serves as a major HS-binding site for Shh (Carrasco et al., 2005; Rubin et al., 2002). This sequence is necessary for Shh binding to HS.

Concerning other factors and in particular FGF signalling, please see response 2 to reviewer 1.

2) If there is an increase in NDST, do the authors have information about any HS core proteins that might also be upregulated?

We have preliminary data concerning this issue. We compared by qRT-PCR the expression level of transcripts of the main proteoglycan HS transcripts in mouse CC seven days after injection of the LPC and showed that only glypicans 4 and 5 are over-expressed after injury. Nevertheless, we would rather not include these data in the work since there are several carriers, Glypicans, Syndecans and perlecans, and since their activity is also depending on the fact that some are membrane bound others are soluble. Here focusing on Ndst1 we work on HS “as a whole” and would rather keep this level of complexity. However, these data can be provided and included in the manuscript as a supplementary figure if requested by the reviewer.

3) Why would a sulfation change in mature oligos increase myelination? Surely OPCs would be more relevant?

As discussed in the manuscript we believe that HS on OLG changes the amount/stability of soluble factors, and thus the recruitment and proliferation of surrounding cells (such as OPC or microglia). Thus, OLG trigger non cell autonomous effects by generating a matrix (environment) that in turn regulates the behavior of cells in the lesioned CC. This point is presented in the Discussion.

4) The authors show that there is some increase in Shh binding around lesions, but not that it does anything. The statement in the Discussion needs to be modified. Also they do not show local upregulation. Any published information that might link Shh or Shh inhibitor to remyelination needs to be stated. This part of the paper needs attention.

The data concerning pattern of expression and role of shh during demyelination and myelin repair are already published and were mentioned in the Discussion of the first version of the manuscript. To be more accurate and better present these data we have now included a sentence in the Introduction “Interestingly oligodendrocyte lineage cells also produce factors that can modulate remyelination like the morphogen Shh which is produced by OLG and OPC at the onset of demyelination in lysophosphatidyl choline (LPC)-induced lesions (Ferent et al., 2013). In this context, blocking Shh activity induces an increase in lesion size and a block in OPC proliferation and differentiation, and conversely Shh overexpression leads to the attenuation of the lesion extent and promotes oligodendrogenesis” These data are also included in the Discussion. See also response 2 to reviewer 1.

5) Do the authors believe that the changes they see in NDST etc. are specific to demyelination, or are similar changes seen with traumatic lesions and inflammation?

As mentioned below and shown in Figure 1—figure supplement 1, we have observed the same pattern of Ndst1 expression in the EAE model. It would indeed be interesting to explore non-demyelinating models in the context of another study.

6) Figure 4: Is the increase in cell proliferation in the knockouts entirely accounted for by increased division of microglia?

We have performed co-immunolabelling of Olig2 and Ki67 showing a significant increase of OPC proliferation mutant mouse compared to control 8dpi. These results are now included in the new version of the manuscript.

7) Figure 6 shows some increase in Shh binding around the lesion, but also widespread binding elsewhere. Some comment is warranted.

Indeed as described in the literature Shh receptors are expressed in various cortical populations in the brain (Harwell et al., Neuron 2012 Mar 22;73(6); Shikawa et al., Dev Biol. 2011 Jan 15;349(2):147-59). As shown in Figure 6B of the presented manuscript AP-SHH probe binds on both size of the brain in the Cortex independently of lesion side.

[Editors’ note: what follows is the authors’ response to the second round of review.]

The authors have submitted a comprehensive revision of the manuscript, with several pieces of new data. These changes address the main issues raised by the reviewers, and the paper is considerably improved. However, the authors have not addressed the mechanistic underpinning of the proposed mechanism – i.e. the belt of HS synthesis and deposition in response to Ndst1 upregulation in oligodendrocytes around the lesion, and the proposal that binding of Shh promotes both, OPC proliferation and microglia polarization. Three points require further revisions.1) We think a fairly simple IHC experiment with existing materials should provide direct mechanistic insight:a) by showing that HS production around the LPC lesions is lost or reduced in the Ndst1 KO

All the requested experiment could not be performed on existing materials since tissue preparation is different depending on the antibodies used; for example, heparan sulfate immunostaining had to be performed on fresh frozen samples while microglia staining requested strong fixative procedure. We had thus to produce new cohorts of mutant mice and perform LPC-induced demyelination to satisfy the reviewer’s requests, explaining the delay in our response.

b) by showing that after LPC lesion the Shh downstream targets in OPCs (and in other cells) are less activated in Ndst1 KO mice

Despite numerous tests, we have reached the conclusion that shh signaling pathway activation cannot be monitored by immunostaining since there are no good antibodies against shh downstream targets. We contacted various experts including Dr Traifford, Dr Zeller and Dr Dahmane who confirmed these points.

We thus used RNAscope technic to perform comparative analysis of patched1 expression in wild-type and mutant animals. We could show that *Ptch1* expression is decreased around LPC-induced demyelination lesions at 8 dpi in mutant animals compared to controls.

These data are now included in a new version of Figure 6 and a paragraph has been added in the Results section.

c) by showing that M2 microglia are indeed increased in number.

We believe that the request is concerning M1 Microglia which is the population we have observed to be affected in mutant condition. We gained information on microglia phenotype using Cox2 immunostaining, Cox2 being a pro-inflammatory enzyme thus labeling M1 microglia. We show a significant increase of Cox2+ cells /mm2 in Ndst1 KO mice 8 days post lesion indicating that the number of M1 microglia is increased in mutant mice after demyelination.

2) The authors are also encouraged to check for Ndst1 upregulation in the available databases on single nuclei gene expression in MS tissue.

Interrogation of our Jaekel et al., 2019 single nuclei RNAseq dataset shows few nuclei that express NDST1 RNAs generally, which is likely due to the dropseq platform only identifying around 15% of the nuclear RNAs, and so favors RNAs with high expression. This is also very similar on interrogation of the Rowitch dataset from human brain.

Therefore, there are too few nuclei here expressing NDST1 to be able to be confident. However, with this caveat, if the cells that express NDST1 are compared, there is a trend to increased expression in MS tissue, which also holds true if only oligodendroglia expressing NDST1 are compared, supporting our biological conclusions. We added part of these data to the manuscript as Figure 8—figure supplement 1.

3) Finally, the role of microglia has been experimentally explored by additional experiments, but the possibility of non-cell autonomous effects on remyelination (by microglia) has not been adequately discussed, despite some reassurance in the rebuttal letter.

A paragraph has been added in the Discussion.